

# An eleven year record of XCO₂ estimates derived from GOSAT measurements using the NASA ACOS version 9 retrieval algorithm

Thomas E. Taylor[1], Christopher W. O'Dell[1], David Crisp[2], Akhiko Kuze[3], Hannakaisa Lindqvist[4], Paul O. Wennberg[5], Abhishek Chatterjee[6,7], Michael Gunson[2], Annmarie Eldering[2], Brendan Fisher[2], Matthäus Kiel[2], Robert R. Nelson[2], Aronne Merrelli[8], Greg Osterman[2], Frédéric Chevallier[9], Paul I. Palmer[10], Liang Feng[10], Nicholas M. Deutscher[11], Manvendra K. Dubey[12], Dietrich G. Feist[13,14,15], Omaira E. García[16], David W. T. Griffith[11], Frank Hase[17], Laura T. Iraci[18], Rigel Kivi[19], Cheng Liu[20], Martine De Mazière[21], Isamu Morino[22], Justus Notholt[23], Young-Suk Oh[24], Hirofumi Ohyama[22], David F. Pollard[25], Markus Rettinger[26], Coleen M. Roehl[27], Matthias Schneider[26], Mahesh Kumar Sha[28], Kei Shiomi[3], Kimberly Strong[29], Ralf Sussmann[26], Yao Té[30], Voltaire A. Velazco[11,31], Mihalis Vrekoussis[32,23], Thorsten Warneke[23], and Debra Wunch[29]

[1]Cooperative Institute for Research in the Atmosphere, Colorado State University, Fort Collins, CO, USA.
[2]Jet Propulsion Laboratory, California Institute of Technology, Pasadena, CA, USA.
[3]Japan Aerospace Exploration Agency, Tsukuba-City, Ibaraki, Japan
[4]Finnish Meteorological Institute, Helsenki, Finland
[5]California Institute of Technology, Pasadena, CA, USA.
[6]Universities Space Research Association, Columbia, MD, USA
[7]Goddard Space Flight Center, Greenbelt, MD, USA
[8]Space Science and Engineering Center, University of Wisconsin - Madison, Madison WI, 53706, USA.
[9]Laboratoire des Sciences du Climat et de l'Environnement/IPSL, CEA-CNRS-UVSQ, Université Paris-Saclay, 91198 Gif-sur-Yvette, France
[10]National Centre for Earth Observation, University of Edinburgh, Edinburgh, UK
[11]Centre for Atmospheric Chemistry, School of Earth, Atmospheric and Life Sciences, University of Wollongong, Wollongong, Australia
[12]Los Alamos National Laboratory, Los Alamos, NM 87545, USA
[13]Max Planck Institute for Biogeochemistry, Jena, Germany
[14]Ludwig-Maximilians-Universität München, Lehrstuhl für Physik der Atmosphäre, Munich, Germany
[15]Deutsches Zentrum für Luft- und Raumfahrt, Institut für Physik der Atmosphäre, Oberpfaffenhofen, Germany
[16]Izaña Atmospheric Research Centre (IARC), State Meteorological Agency of Spain (AEMET), Santa Cruz de Tenerife, Spain
[17]Karlsruhe Institute of Technology, IMK-ASF, Karlsruhe, Germany
[18]NASA Ames Research Center, Moffett Field, CA, USA
[19]Finnish Meteorological Institute, FMI, Sodankylä, Finland
[20]Department of Precision Machinery and Precision Instrumentation, University of Science and Technology of China, Hefei 230026, China
[21]Royal Belgian Institute for Space Aeronomy (BIRA-IASB), Brussels, Belgium
[22]National Institute for Environmental Studies (NIES), Tsukuba, Japan
[23]Institute of Environmental Physics, University of Bremen, Bremen, Germany
[24]Global Atmosphere Watch Team, Innovative Meteorological Research Department, National Institute of Meteorological Sciences, Jeju-do, Republic of Korea
[25]National Institute of Water and Atmospheric Research Ltd (NIWA), Lauder, New Zealand
[26]Karlsruhe Institute of Technology, IMK-IFU, Garmisch-Partenkirchen, Germany
[27]Division of Geological and Planetary Sciences, California Institute of Technology, Pasadena, CA, USA





[28]SRON Netherlands Institute for Space Research, Utrecht, the Netherlands
[29]Department of Physics, University of Toronto, Toronto, Ontario, Canada
[30]Laboratoire d'Etudes du Rayonnement et de la Matière en Astrophysique et Atmosphères (LERMA-IPSL), Sorbonne
Université, CNRS, Observatoire de Paris, PSL Université, Paris, France
[31]Deutscher Wetterdienst, Meteorological Observatory Hohenpeissenberg, 82383 Germany
[32]Climate and Atmosphere Research Center (CARE-C), The Cyprus Institute, Nicosia, Cyprus

**Correspondence:** Tommy Taylor (tommy.taylor@colostate.edu)

**Abstract.** The Thermal And Near infrared Sensor for carbon Observation - Fourier Transform Spectrometer (TANSO-FTS) on the Japanese Greenhouse gases Observing SATellite (GOSAT) has been returning data since April 2009. The version 9 (v9) Atmospheric Carbon Observations from Space (ACOS) Level 2 Full Physics (L2FP) retrieval algorithm (Kiel et al., 2019) was used to derive estimates of carbon dioxide ($CO_2$) dry air mole fraction ($XCO_2$) from the TANSO-FTS measurements collected

5  over it's first eleven years of operation. The bias correction and quality filtering of the L2FP $XCO_2$ product were evaluated using estimates derived from the Total Carbon Column Observing Network (TCCON) as well as values simulated from a suite of global atmospheric inverse modeling systems (models). In addition, the v9 ACOS GOSAT $XCO_2$ results were compared with collocated $XCO_2$ estimates derived from NASA's Orbiting Carbon Observatory-2 (OCO-2), using the version 10 (v10) ACOS L2FP algorithm.

These tests indicate that the v9 ACOS GOSAT $XCO_2$ product has improved throughput, scatter and bias, when compared to the earlier v7.3 ACOS GOSAT product, which extended through mid 2016. Of the 37 million (M) soundings collected by GOSAT through June 2020, approximately 20% were selected for processing by the v9 L2FP algorithm after screening for clouds and other artifacts. After post-processing, 5.4% of the soundings (2M out of 37 M) were assigned a "good" $XCO_2$ quality flag, as

15  compared to 3.9% in v7.3 (<1 M out of 24 M). After quality filtering and bias correction, the differences in $XCO_2$ between ACOS GOSAT v9 and both TCCON and models have a scatter (one sigma) of approximately 1 ppm for ocean-glint observations and 1 to 1.5 ppm for land observations. Similarly, global mean biases are less than approximately 0.2 ppm. Seasonal mean biases relative to the v10 OCO-2 $XCO_2$ product are of order 0.1 ppm for observations over land. However, for ocean-glint observations, seasonal mean biases relative to OCO-2 range from 0.2 to 0.6 ppm, with substantial variation in time and latitude.

The ACOS GOSAT v9 $XCO_2$ data are available on the NASA Goddard Earth Science Data and Information Services Center (GES-DISC) (OCO-2 Science Team et al., 2019a, b). The v9 ACOS Data User's Guide (DUG) describes best-use practices for the data (O'Dell et al., 2020). This dataset should be especially useful for studies of carbon cycle phenomena that span a full decade or more, and may serve as a useful complement to the shorter OCO-2 v10 dataset, which begins in September 2014.





## 1 Introduction

A new era of dedicated satellite observations of greenhouse gases began in 2009, with the successful launch of GOSAT (Kuze et al., 2009). Each day, GOSAT's TANSO-FTS acquires approximately ten thousand high spectral resolution measurements of reflected sunlight ($\simeq 36.5$ M in ten years). Soundings that are determined to be sufficiently clear of clouds and aerosols are processed by retrieval algorithms to produce estimates of $XCO_2$ and column-average dry air mole fractions of methane ($CH_4$). Both the quality of the GOSAT TANSO-FTS spectra and the derived $XCO_2$ and $XCH_4$ estimates have been continually refined over the past twelve years. While the official GOSAT L2 products are available from the National Institute for Environmental Studies (NIES; http://www.gosat.nies.go.jp/en/about_5_products.html; Yoshida et al., 2013) a number of independent research institutes have developed their own products e.g., Butz et al., 2011; Crisp et al., 2012; Cogan et al., 2012; Heymann et al., 2015; Parker et al., 2020.

One of these groups, the Atmospheric $CO_2$ Observations from Space (ACOS) team, used an L2FP retrieval algorithm developed for the NASA Orbiting Carbon Observatory (OCO) to derive estimates of $XCO_2$ from the GOSAT data (O'Dell et al., 2012; Crisp et al., 2012). Early $XCO_2$ estimates from these efforts had large biases and random errors when compared to $XCO_2$ estimates from the Total Carbon Column Observing Network (TCCON) and other standards. For example, the v2.8 ACOS GOSAT L2FP product had biases of 7 to 8 ppm relative to TCCON (Crisp et al., 2012). These biases were reduced to 1-2 ppm in the v2.9 product. The next major release was v3.5 in 2014, which spanned approximately 4 years. This data product showed additional reductions in bias and scatter against TCCON, as well as reasonable agreement in seasonal cycle phase and amplitude (Lindqvist et al., 2015; Kulawik et al., 2016).

These early space-based $XCO_2$ products were rapidly adopted by the carbon cycle science community. Early studies based on GOSAT ACOS retrievals included Basu et al. (2013), Deng et al. (2014), Chevallier et al. (2014), and Feng et al. (2016). These studies provided the first comprehensive insights into regional flux estimates from space-based observations of carbon dioxide. Houweling et al. (2015) conducted an extensive inter-comparison of the early GOSAT-based atmospheric inversion system studies and reported a reduction in the global land sink for $CO_2$ and a shift in the terrestrial net uptake of carbon from the tropics to the extratropics. However, these studies also highlighted the role of spatiotemporal systematic errors in the satellite retrievals and the negative impact they can have on estimation of $CO_2$ sources and sinks using atmospheric inversion systems.

Motivated by these early studies, as well as the launch of the OCO-2 sensor in July 2014, the ACOS team continued to refine the L2FP retrieval. In 2016, the ACOS GOSAT v7.3 product was distributed. No formal results of the $XCO_2$ estimates were published by the algorithm team, although internal analysis showed small improvement over v3.5, as well as an extension of the record to 7 years. A number of atmospheric inversion studies were published using the v7.3 product. For example, Chatterjee et al. (2017) and Liu et al. (2017) used v7.3 to define the climatological background in their studies of the impact of the 2015-2016 El Niño on the tropical carbon cycle. Palmer et al. (2019) used this data product in a global study, concluding that the the
tropical land regions were a net annual source of $CO_2$ emissions, including unexpectedly large net emissions from northern tropical Africa. Wang et al. (2019) found that the ACOS GOSAT v7.3 $XCO_2$ yielded a stronger carbon land sink than the v7 OCO-2 product. Byrne et al. (2020) used the ACOS GOSAT 7.3 product to study interannual variability in the carbon cycle across North America, and Jiang et al. (2021) investigated interannual variability of the carbon cycle across the globe with v7.3.

Most recently, the v9 ACOS L2FP retrieval algorithm, first applied to OCO-2 (Kiel et al., 2019), was used to generate estimates of $XCO_2$ from an eleven year record of GOSAT measurements, spanning April 2009 through June 2020. This both extends the time record over v7.3, and produces an ACOS GOSAT product that is more directly comparable to the newest OCO-2 product, which is now using version 10.

The paper is organized as follows; Section 2 discusses the GOSAT TANSO-FTS instrument and measurements as related to the ACOS $XCO_2$ estimates. In Section 3 updates to the ACOS v9 L2FP algorithm are detailed, and an assessment is given of the v9 $XCO_2$ data product volume. The $XCO_2$ quality filtering and bias correction procedures, specific to ACOS GOSAT v9, are also discussed. Section 4 provides an evaluation of the v9 $XCO_2$ product using estimates of $XCO_2$ from TCCON and from a suite of 4 atmospheric inversion systems (models). In addition, a comparison to collocated $XCO_2$ estimates derived from
NASA's OCO-2 sensor is presented. A summary of the results is provided in Section 5.

## 2   The GOSAT instrument and measurements

The GOSAT mission is a joint project between the Japan Aerospace Exploration Agency (JAXA), the National Institute for Environmental Studies (NIES), and the Ministry of the Environment (MOE) (Kuze et al., 2009). GOSAT was launched on 23
January 2009 into a sun-synchronous orbit with a local overpass time of approximately 12:49PM and a 3 day ground repeat cycle. Its TANSO-FTS collects high resolution spectra of reflected sunlight that can be analyzed to yield estimates of the greenhouse gases carbon dioxide ($CO_2$) and methane ($CH_4$) (Yoshida et al., 2011, 2013).

### 2.1   GOSAT TANSO-FTS instrument

TANSO-FTS collects high resolution spectra of reflected sunlight in the near infrared (NIR) and short-wave infrared (SWIR) spectral ranges that include the the oxygen A-band near 0.76 $\mu$m (ABO2 band) at approximately $0.36\,\mathrm{cm}^{-1}$ spectral resolution, and weak and strong $CO_2$ absorption features near 1.6 $\mu$m (WCO2 band) and 2.0 $\mu$m (SCO2 band), respectively, at $0.27\,\mathrm{cm}^{-1}$ spectral resolution. All three channels simultaneously measure two orthogonal components of polarization approximately every 4.6 seconds.




Each GOSAT sounding has a circular ground footprint with a diameter of approximately 10.5 km when viewing the local nadir. An agile, two-axis pointing system allows cross-track and along-track motions of $\pm 35°$ and $\pm 20°$, respectively. Before August 2010, a 5-point cross-track scan was used, yielding footprints that were separated by approximately 150 km in both the down-track and along-track dimensions. Since that time, a 3-point cross-track scan has been used, yielding footprint separation of approximately 260 km (Kuze et al., 2016).

Over water, the TANSO-FTS scan mechanism targets the field of view to collect observations in the direction of the local glint spot, where sunlight is specularly reflected from the surface. Early in the mission, glint observations were collected only within $\pm 20°$ of the sub-solar latitude. In May 2013, to increase the latitudinal extent of the GOSAT Ocean-Glint measurements, the scanning strategy was improved to better track the actual specular glint spot, which varies by latitude and season. The latitude range for glint observation was further extended three times in increments of $3°$ in September 2014, June 2015, and January 2016, by not only tracking the exact specular point but also tracking along the principal plane of the specular reflection when the glint spot was out of range of the scan mechanism. In addition, more observations over fossil fuel emission target sites such as mega-cities and power plants have been made in recent years, allowing for detailed emission source studies (e.g., Kuze et al., 2020). Daily observation patterns can be found at https://www.eorc.jaxa.jp/GOSAT/currentStatus_10.html.

The TANSO-FTS detectors can be read out using independent medium-gain and high-gain signal chains. Most measurements over land use the instrument's high-gain signal chain (Land H-gain), while brighter land surfaces are measured using the medium-gain signal chain (Land M-gain) to avoid saturating the detectors. Over oceans, which appear dark in the SWIR spectral bands, measurements are collected using the high-gain signal chain (Ocean H-gain or Ocean-Glint) to maximize the signal.

During the first 7 years of GOSAT operations (2009-2015), data acquisition was temporarily suspended due to one spacecraft and two instrument anomalies, as highlighted in Kuze et al. (2016). A rotation failure of a solar paddle in 2014 resulted in a data loss of 6 days. A switch from the primary to secondary pointing mirror in January 2015 resulted in a data loss of approximately 6 weeks, while a temporary shutdown of the cryocooler in August 2015 resulted in a data loss of 13 days.

Since 2015, three additional anomalies interrupted data acquisition. An unexpected shutdown of the instrument occurred in May 2018, resulting in the loss of a week of data. A failure of the second solar panel caused a significant loss of data spanning more than a month in November and December of 2018, and an anomaly of the FTS alignment laser, caused a loss of a week of data in June of 2020. In all these cases, the system was able to recover full functionality either through utilization of on-board back-up systems, or through mitigation strategies, and as of the summer of 2021, TANSO-FTS continues to collect science data.





## 2.2 ACOS GOSAT v9 L1b measurements

The JAXA L1b algorithm, which has been updated more than 10 times over the eleven year data record, produces an internally
consistent set of geometrically, radiometrically, and spectrally calibrated TANSO-FTS radiances. The raw spectral measurements are interferograms, which are calibrated and Fourier transformed to yield spectra. The version 205/210 Level 1b (L1b) geolocated and calibrated radiances provided by JAXA have been used for the ACOS v9 reprocessing. A list of L1b updates for v205/210 can be found in Table 3 of the ACOS v9 Data Users Guide (DUG) (O'Dell et al., 2020). Note that while the current L1b version is now 230, the only differences between this version and 205/210 are in the thermal infrared band (5.6 -
14.3 $\mu$m), which is not used in the ACOS $XCO_2$ retrieval.

After obtaining the calibrated L1b product from JAXA, the ACOS team converts the files to the format needed as input to the ACOS L2 algorithms. The L2FP algorithm uses a simple average of the S and P linear polarizations to produce an approximation of the total measured intensity. Due to cooperation agreements between JAXA and the California Institute of Technology,
the distribution of the ACOS GOSAT L1b product is restricted and therefore not publicly available on the NASA DISC. However, the data may be procured by submitting a request to the GOSAT project.

## 3    The ACOS v9 L2FP $XCO_2$ retrieval algorithm

The ACOS Level 2 full physics (L2FP) retrieval algorithm is well documented, most recently in O'Dell et al. (2018) for v8
and in Kiel et al. (2019) for v9. A Bayesian optimal estimation framework is used to derive estimates of $XCO_2$ from spectral measurements of reflected solar radiation. A post-processing step assigns a simple good/bad quality flag (QF) to each $XCO_2$ value based on successful L2FP algorithm convergence and a series of empirically derived filters. An empirical bias correction (BC) to the estimated $XCO_2$ values, derived from comparisons with TCCON derived $XCO_2$ and $CO_2$ fields from a suite of atmospheric inversion systems, is included in the Lite File product. Here we provide a summary of the recent evolution of the
ACOS algorithm and discuss retrieval parameters and setup specific to GOSAT.

### 3.1    ACOS L2FP algorithm updates

Table 1 summarizes the evolution of the ACOS L2FP retrieval algorithm from v7 to v10. A similar table, complete through v8, can be found in O'Dell et al. (2018). The trace gas absorption coefficient tables (ABSCO) were updated from v4.2 (Thompson
et al., 2012) in ACOS v7 to ABSCO v5.0 (Oyafuso et al., 2017) in ACOS v8/9. The ACOS v9 L2FP algorithm is unmodified relative to v8 (Kiel et al., 2019). However, changes were made in v9 regarding the sampling of the meteorological prior, which does affect ACOS GOSAT estimates of $XCO_2$. The source of the prior meteorology was switched from the European Center for Medium-range Weather Forecast (ECMWF) in ACOS v7, to the NASA Goddard Modeling and Assimilation Of-





**Table 1.** Updates to recent versions of the ACOS L2FP retrieval algorithm. N/C stands for No Change.

|   |   | ACOS v7 | ACOS v8/v9 | ACOS v10 |
|---|---|---|---|---|
| 1 | Spectroscopy | ABSCO v4.2 | ABSCO v5.0 | ABSCO v5.1 |
| 2 | Meteorology prior source | ECMWF | GEOS5 FP-IT | N/C |
| 3 | Aerosol prior source | MERRA monthly climatology | N/C | GEOS5 FP-IT with tightened prior uncertainty |
| 4 | Retrieved aerosol types | water + ice + 2 MERRA types | + stratospheric aerosol | N/C N/C |
| 5 | AOD prior value (per type) | 0.0375 | 0.0125 | N/C |
| 6 | $CO_2$ prior source | TCCON ggg2014 | N/C | TCCON ggg2020 |
| 7 | Land surface model | Lambertian | BRDF | N/C |

fice (GMAO) Goddard Earth Observing System (GEOS) Forward Processing - Instrument Team (FP-IT) product for ACOS

v8/9. Both v7 and v8/9 used aerosol priors based on a simple monthly 1° latitude by 1° longitude climatology constructed from the output aerosol fields of the GMAO Modern-Era Retrospective analysis for Research and Applications (MERRA) product (Rienecker et al., 2011). However, between v7 and v8/9, an additional stratospheric aerosol layer was introduced, as described in Section 3.1.1 of O'Dell et al. (2018). In addition, the prior value of the aerosol optical depth (AOD) for each retrieved aerosol type was lowered from 0.0375 in ACOS v7 to 0.0125 in ACOS v8/9 based on extensive testing. There was

no change in the source of the $CO_2$ prior from ACOS v7 to v8/9; both versions adopted the prior developed by the TCCON team for use in the ggg2014 algorithm (Wunch et al., 2015). An additional change from ACOS v7 to v8/9 was a switch from a purely Lambertian land surface model, to a more sophisticated bi-directional reflectance distribution function (BRDF) model.

Several important components of the v9 ACOS L2FP retrieval configured for GOSAT have not changed from v7.3; (i) the sur-

face pressure prior constraint remains set at ±2 hPa, (ii) three Empirical Orthogonal Functions (EOFs) are fit in each spectral band (see Section 3.3 in O'Dell et al. (2018) for a full discussion of ACOS EOFs), and (iii) a zero level offset (ZLO) is fit in the state vector to account for non-linearity in the ABO2 signal chain on GOSAT TANSO-FTS (Crisp et al., 2012).

To support comparisons of the ACOS GOSAT v9 $XCO_2$ product with the OCO-2 v10 product, Table 1 includes the most recent

updates to the ACOS v10 L2FP algorithm. For v10, the ABSCO tables were again updated from v5.0 to v5.1 (Payne et al., 2020). The aerosol prior was updated from the MERRA monthly climatology to daily GEOS-FT-IT values, with a tightened prior uncertainty (Nelson and O'Dell, 2019). Finally, the $CO_2$ priors developed by the TCCON team for use in ggg2014 were updated to a revised set of priors developed for use in ggg2020.



**Table 2.** Accounting of the soundings in the eleven year long GOSAT ACOS v9 dataset at each stage of the data processing chain. The final line summarizes the number of good quality XCO$_2$ soundings used in the evaluation section of this work.

| | Number of soundings (N) | Total (%) | Fraction of Selected (%) | Valid (%) |
|---|---|---|---|---|
| Total in ACOS GOSAT v9 record | 37.4 M | 100. | – | – |
| Selected for L2FP | 7.0 M | 18.8 | 100. | – |
| Non-convergence (terminated at unphysical state) | 0.3 M | 0.7 | 3.9 | – |
| Non-convergence (exceeded iteration limit) | 0.2 M | 0.6 | 3.2 | – |
| Non-convergence (exceeded diverging steps limit) | 0.4 M | 1.1 | 5.9 | – |
| Valid, converged L2FP XCO$_2$ result | 6.1 M | 16.4 | 87.0 | 100. |
| Lite file aggregator IDP filtering | 0.3 M | 0.7 | 3.9 | 4.5 |
| Bad L2Lite quality flag | 3.9 M | 10.4 | 55.3 | 63.6 |
| Good L2Lite quality flag | 2.0 M | 5.4 | 28.6 | 32.9 |

### 3.2 ACOS GOSAT v9 L2FP sounding selection and convergence

GOSAT data from April 20, 2009 through June 30, 2020 were passed through the ACOS L2FP algorithm pipeline, which includes a series of stages where soundings can be rejected or selected for further processing. The throughput of each of these stages for ACOS GOSAT v9 is summarized in Table 2 and Figure 1. The pipeline begins with a series of preprocessing steps, which reject corrupted spectra and screen the remainder to eliminate those with optically-thick clouds and/or aerosols (Taylor et al., 2016). From the full set of measurements (Panel A of Figure 1), the remaining soundings are accepted by the L2FP algorithm (18.8% of the 37.4 M measured soundings contained in the ACOS GOSAT v9 record) (Panel B of Figure 1) and a retrieval of XCO$_2$ is attempted. The majority of the selected soundings successfully converge to a valid solution; 87% for ACOS GOSAT v9 (16.4% of the total measured soundings). Soundings can fail to converge for a variety of reasons, including (i) producing non physical values, such as negative gas mixing ratios or surface pressures (3.9% of the selected), (ii) converging too slowly and exceeding a predefined number of iterations (3.2% of the selected), or (iii) having more diverging steps than the predefined maximum (5.9% of the selected). The 6.1 M valid soundings were then run through the quality filtering and bias correction procedure discussed in the next section.

### 3.3 ACOS GOSAT v9 XCO$_2$ quality filtering and bias correction

All GOSAT soundings that converged to a valid XCO$_2$ value within the L2FP retrieval were input to the quality filtering and bias correction procedure. A modest fraction (4.5% of the valid soundings) were removed from the final L2Lite product based on screening via the IDP CO$_2$ ratio, which indicated the presence of clouds or aerosols. Based on a series of screening criteria

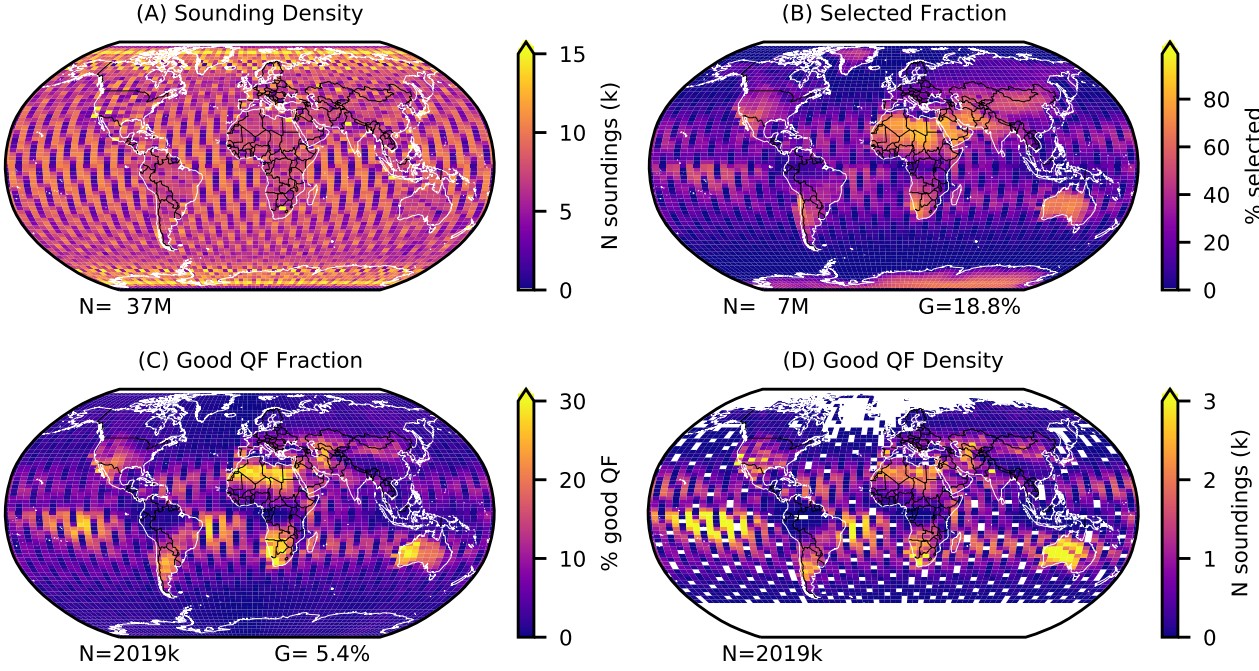

**Figure 1.** The total measured sounding density per 2.5° by 5° latitude/longitude grid cell in the eleven year (April 2009 - June 2020) ACOS GOSAT v9 data record (A). The fraction of the total soundings selected to run through the L2FP algorithm (B). The fraction of the total soundings that converged in the L2FP and were assigned a good L2FP QF (C). The sounding density of the good QF data per 2.5° by 5° latitude/longitude grid cell (D).

derived from comparisons with TCCON and modeled $CO_2$ fields, each sounding that converged within the L2FP is assigned either a "good" (=0) or "bad" (=1) $XCO_2$ quality flag. Generally, for global or regional studies, it is recommended that users
retain only the "good" quality soundings, as the soundings flagged as "bad" quality are likely to include biases that compromise their utility for some applications. A global map of the ACOS GOSAT v9 "good" $XCO_2$ sounding density is provided in panels C and D of Figure 1. A subset of data variables from the per-orbit L2Std files (OCO-2 Science Team et al., 2019b), along with the quality filter flag and bias corrected $XCO_2$, are repackaged into the daily aggregated L2Lite NetCDF files (OCO-2 Science Team et al., 2019a).


A fundamental aspect of the quality filtering and bias correction procedures (QF/BC) is the need for $XCO_2$ truth metrics with which to compare the satellite derived estimates (O'Dell et al., 2018). The development of ACOS GOSAT v9 used $XCO_2$ truth metrics derived from both TCCON measurements, and the median $CO_2$ distributions determined from a suite of four atmospheric inversion systems.




TCCON is a well established validation transfer standard for space-based estimates of $XCO_2$ (Wunch et al., 2011a, 2017b). For the ACOS GOSAT v9 QF/BC, estimates of $XCO_2$ derived from TCCON measurements using the ggg2014 retrieval algorithm were used (Wunch et al., 2015). Individual GOSAT soundings were compared to TCCON daily mean $XCO_2$ values. TCCON data were included if: (i) they were flagged good (flag = 0), (ii) they fell within 3 standard deviations of a daily quadratic fit
against time (to remove outliers, e.g. due to unscreened cloud), (iii) they covered at least 15 minutes within a given day, (iv) there were at least 3 good soundings within the day, and (v) the standard deviation of the good soundings for the day was less than 3 ppm. In the GOSAT-TCCON comparisons described here, an averaging kernel correction was applied to each TCCON $XCO_2$ estimate following Nguyen et al. (2014), prior to calculating the daily mean value.

The default spatial collocation criteria for GOSAT soundings were those falling within $\pm 2.5°$ latitude and $\pm 5°$ longitude of a TCCON station for most sites. For southern hemisphere (SH) sites poleward of $25°$ S latitude, where the variation of $CO_2$ is low, the spatial box was increased to $\pm 10°$ latitude by $20°$ longitude to increase the number of collocations. For the Edwards TCCON station, which lies in an arid region just north of the polluted Los Angeles metropolitan area, a very specific colloca-tion box of [34.68, 37.46] latitude and [-127.88,-112.88] longitude was used to avoid contamination from the city. Similarly,
for the Caltech site, located in Pasadena, California, a latitude box of [33.38, 34.27] and longitude box of [-118.49, -117.55] was used. This avoids collocating GOSAT soundings measured over ocean, the San Gabriel Mountains, and regions too far outside of the Los Angeles basin with the Caltech TCCON data. Finally, only GOSAT soundings acquired within $\pm 2$ hours of the mean TCCON measurement time were considered. For the quality filtering and bias correction procedure, single sounding level collocations are used to maximize the number of fit points.


Estimates of $CO_2$ from atmospheric inversion systems, or models, provide a useful metric for evaluating satellite based esti-mates of $XCO_2$ (O'Dell et al., 2018). In this work, a suite of four models (CarbonTracker, CAMS, CarboScope, and Univ. of Edinburgh) were sampled at the GOSAT sounding times and locations. Brief descriptions of each, along with references, are provided in Table 3. The models use a variety of land biosphere prior fluxes, inverse solvers and transport models, and assim-
ilate $CO_2$ data only from flasks and continuous analyzers on a wide variety of platforms, e.g., observatories, towers, aircraft, and ships. Specifically, no data from GOSAT, OCO-2, or TCCON are assimilated. The $CO_2$ concentration fields of the models capture the known features of the global atmospheric $CO_2$ distribution, including seasonality, time trends and inter-annual variability (IAV) due to ENSO. For each GOSAT sounding, the vertical profiles of $CO_2$ from the corresponding grid box of each of the four models are spatiotemporally interpolated (linear in latitude, longitude, and time) to the GOSAT observation
point, and the GOSAT averaging kernel is applied to each vertical profile to produce a modeled $XCO_2$ as if viewed from the satellite.

For each GOSAT sounding, a multi-model median (MMM) $XCO_2$ was calculated from the models having a valid $XCO_2$ es-timate for that location and time. To exclude outliers, models with $XCO_2$ that deviated more than $\pm 1.5$ ppm from the initial
MMM for that sounding were not included. The sounding was then rejected if more than one of the four models had been





**Table 3.** Carbon inversion systems used for ACOS GOSAT v9.

| Model Name | Institute | Land Biosphere Prior | Transport Model | Inverse Method | Citations |
|---|---|---|---|---|---|
| CarbonTracker | NOAA Global Monitoring Laboratory | CASA | TM5 | EnKF | Peters et al. (2007) CarbonTracker (2021) |
| CarboScope | Max Planck Institute for Biogeochemistry | Zero | TM3 | 4D-Var | Rödenbeck (2005); Rödenbeck et al. (2018) CarboScope (2021) |
| CAMS | Copernicus Atmosphere Monitoring Service | ORCHIDEE | LMDZ | 4D-Var | Chevallier et al. (2010) CAMS (2021) |
| UoE | University of Edinburgh Atmospheric Composition Modelling Group | CASA | GEOS-Chem | EnKF | Feng et al. (2009) UoE (2021) |

**Table 4.** Carbon inversion system data sets used for the QF/BC and $XCO_2$ evaluation of ACOS GOSAT v9.

| Model | QF/BC | Evaluation |
|---|---|---|
| CarbonTracker | CT2017 (through 20170429) CT-NRT.v2019-2 (through 20190330) | CT2019 (through 20181231) |
| CarboScope | Jena_s04c_v4.3 | Jena_s10oc-v2020 |
| CAMS | v18r2 (second release of CAMS data that extends through 2018) | v20r1 (first release of CAMS data that extends through 2020) |
| UoE | v4.0 (used in Palmer et al. (2019)) | v4.0a (an extension of v4.0, using near real-time in situ data for 2019. |

excluded, or if the standard deviation amongst the valid models was >1 ppm. Approximately 90% of the GOSAT v9 soundings had a valid MMM $XCO_2$ value for analysis. Table 4 lists the model version numbers used for the QF/BC procedure, as well as that used in the evaluation of the final good quality $XCO_2$ product that will be presented later.

Table 5 lists the quality filtering variables used for ACOS GOSAT v9 and their corresponding thresholds. Many of the same variables (18 out of 31) were also used in the OCO-2 v9 quality filtering, as seen in Table 5 of Kiel et al. (2019). This includes the IMAP-DOAS Preprocessor (IDP) $CO_2$ and $H_2O$ ratios (Frankenberg et al., 2005), and the A-Band Preprocessor $dP$, i.e., the difference between the retrieved and prior surface pressure from the Oxygen-A band (Taylor et al., 2016). Another common variable used for quality filtering is the perturbation in the L2FP $CO_2$ vertical profile relative to the prior, a quantity called

"$CO_2$ grad del" ($\delta \nabla_{CO_2}$), as defined in equation 5 of O'Dell et al. (2018). A number of aerosol related retrieval parameters are also used, similar to OCO-2 v9. Section 2.5 of the ACOS GOSAT v9 DUG provides additional details on the quality filtering





(O'Dell et al., 2020).

Spurious correlations in the estimates of $XCO_2$ with other retrieval variables due to inadequacies in the modeled physics mo-
tivate the application of a bias correction (Wunch et al., 2011b; O'Dell et al., 2018). Generally such spurious correlations are
found with state vector elements such as retrieved surface pressure, various aerosol parameters, and $\delta \nabla_{CO_2}$. A general discus-
sion of the ACOS $XCO_2$ bias correction methodology is provided in Section 4 of O'Dell et al. (2018).

For interested readers, the explicit formula for application of the correction is provided in Section 2.5.6 of the ACOS GOSAT
DUG (O'Dell et al., 2020). For both land H-gain and M-gain, a set of five BC variables are used, while Ocean-Glint uses only
3 variables. The difference between the H- and M-gain bias correction over land is minor. New for ACOS GOSAT v9 is the
use of a correction against time, which is made possible with an eleven year data record; the corrections are +0.05 ppm/yr over
land and +0.10 ppm/yr over water. The source of this spurious drift in the bias-corrected $XCO_2$ is currently unclear and is the
subject of on going study. Although there is some commonality in the quality filtering and bias correction variables used for
ACOS GOSAT v9 (compare Tables 5 and 6), they do differ somewhat, as is typically the case with each sensor and data version.

Table 6 compares the bias correction variables used for ACOS GOSAT v9 with the variables used in the previous ACOS
GOSAT v7.3, as well as with OCO-2 v9 and v10. The same few variables have appeared in all recent versions, including L2FP
$\delta \nabla_{CO_2}$, L2FP $dP$, and L2FP DWS for land soundings. For ocean soundings the bias correction variables have evolved, with
the only common one being $\delta \nabla_{CO_2}$.

Table 7 summarizes the effect of the quality filtering and bias correction on the ACOS GOSAT $XCO_2$ for v7.3 and v9. For
Ocean-Glint soundings, the v9 quality flag is substantially more restrictive compared to v7.3, i.e. $\simeq 57\%$ pass rate compared
to $\simeq 78\%$. This is mostly driven by the more extensive latitudinal coverage in the v9 record, which tends to include more
soundings with high solar zenith angles (SZA) and low signal to noise ratio (SNR), which are more challenging for the L2FP.
For H-gain land observations, the two versions have quite similar QF pass rates ($\simeq 35$-$45\%$). The QF pass rate for v9 M-gain
Land data is $\simeq 39\%$ when compared against models, but $\simeq 56\%$ against TCCON. In all cases there is a significant reduction in
the scatter of the $XCO_2$ after application of the QF/BC; by a factor of $\simeq 2$ for Ocean-Glint and Land M-gain, and a factor of
3 for Land H-gain. The QF/BC scatter is always slightly lower for v9 compared to v7.3, although the number of soundings is
greater by 1.5 to 10 times for the various scenarios.





**Table 5.** ACOS GOSAT v9 L2FP quality filtering variables and thresholds. Descriptions of the variables can be found in the DUG (O'Dell et al., 2020). Soundings falling outside of the data ranges are assigned a bad $XCO_2$ quality flag. The second column identifies variables that were also used for OCO-2 v9 quality filtering, as taken from Table 5 of Kiel et al. (2019).

| Variable | Used for OCO-2 v9 | Ocean-Glint | Land H-gain | Land M-gain |
|---|---|---|---|---|
| Geo altitude $\sigma$ | Y | NA | $< 250.0$ | $< 250.0$ |
| Geo airmass | N | $< 3.0$ | – | – |
| L1b SCO2/WCO2 signal ratio | N | $> 0.58$ | – | – |
| IDP $CO_2$ ratio | Y | [0.989, 1.02] | [0.95, 1.02] | [0.989, 1.012] |
| IDP $H_2O$ ratio | Y | – | [0.80, 1.04] | [0.88, 1.05] |
| ABP $dP$ (retrieved - prior $p_{surf}$) | Y | [-25.0, 14.0] | [-7.0, 7.0] | [-10.0, 7.0] |
| L2FP outcome flag | Y | 1 or 2 | 1 or 2 | 1 or 2 |
| L2FP Total AOD | Y | $< 0.5$ | [0.02, 0.3] | $< 0.4$ |
| L2FP AOD ice cloud | Y | $< 0.07$ | $< 0.06$ | [0.002, 0.05] |
| L2FP AOD sulfate aerosol | N (used organic carbon) and sea salt, independently) | – | $< 0.20$ | – |
| L2FP AOD stratospheric aerosol | Y | – | – | [0.0008, 0.015] |
| L2FP AOD DWS (dust + sea salt + water cloud) | Y | – | – | [0.0001, 0.35] |
| L2FP AOD fine (organic carbon + sulfate aerosol) | N | $< 0.18$ | – | $< 0.04$ |
| L2FP ice cloud pressure height | Y | [-0.50, 0.40] | [-0.12, 0.40] | [-0.12, 0.30] |
| L2FP dust aerosol pressure height | N | – | [0.75, 1.4] | [0.80, 1.4] |
| L2FP $XCO_2$ uncertainty | Y | – | $< 2.0$ | $< 1.5$ |
| L2FP $CO_2$ grad del ($\delta \nabla_{CO_2}$) | Y | [-19.0, 10.0] | [-40.0, 100.0] | [-10.0, 100.0] |
| L2FP $dP$ (retrieved - prior $p_{surf}$) | Y | [-0.75, 5.5] | [-2.0, 10.0] | [-6.0, 5.0] |
| L2FP WCO2 albedo | N | [0.017, 0.030] | – | – |
| L2FP SCO2 albedo | Y | – | [0.04, 1.0] | [0.10, 1.0] |
| L2FP ABO2 albedo slope | N | – | [-4E-5, 2E5] | – |
| L2FP WCO2 albedo slope | Y | [3E-5, 2.7E-5] | – | – |
| L2FP SCO2 albedo slope | Y | [0.0, 5E-5] | [-1E-4, 2.5E-4] | – |
| L2FP ABO2 $\chi^2$ | N | – | $< 1.2$ | $< 1.25$ |
| L2FP WCO2 $\chi^2$ | Y | $< 1.4$ | – | $< 1.4$ |
| L2FP SCO2 $\chi^2$ | N | $< 1.35$ | – | $< 1.9$ |
| L2FP ABO2 offset | N | – | – | [-1.5, 0.1] |
| L2FP temperature offset | N | – | [-1.0, 10.0] | [-0.7, 1.2] |
| L2FP ABO2 EOF 3 scaling | N | [-0.05, 0.04] | – | – |
| L2FP SCO2 EOF 2 scaling | N (used EOF 3) | [-0.15, 0.35] | – | – |
| L2FP wind speed | Y | [2.0, 24.0] | – | – |





**Table 6.** ACOS L2FP bias correction variables by sensor and product version.

|       | ACOS GOSAT v7.3 | ACOS GOSAT v9 | OCO-2 v9 | OCO-2 v10 | Variable Description |
|-------|-----------------|---------------|----------|-----------|----------------------|
| **Land** | (LandH only) | (LandH/LandM) | (nadir & glint) | (nadir & glint) | |
| | $\delta\nabla_{CO_2}$ | $\delta\nabla_{CO_2}, \delta\nabla_{CO_2}$ | $\delta\nabla_{CO_2}$ | $\delta\nabla_{CO_2}$ | "CO$_2$ grad del" (see text) |
| | $dP$ | – | – | – | Retrieved minus a priori surface pressure |
| | – | dP$_{frac}$, dP$_{frac}$ | dP$_{frac}$ | dP$_{frac}$ | Elevation adjusted $dP$ |
| | | | | | See equation 4 in (Kiel et al., 2019) |
| | DWS | DWS, $\sqrt{(DWS)}$ | DWS | $\log(DWS)$ | Combined aerosol optical depth |
| | | | | | of dust, water, and salt |
| | $\sqrt{\alpha_{SCO_2}}$ | $\alpha_{SCO_2}, \sqrt{\alpha_{SCO_2}}$ | – | – | Square-root of the retrieved |
| | | | | | albedo in the SCO2 spectral band |
| | – | – | – | AOD$_{fine}$ | Combined aerosol optical depth |
| | | | | | of sulfate and organic carbon |
| | – | t$_{year}$ | – | – | Time in years |
| **Ocean** | | | | | |
| | S$_{32}$ | – | – | – | Average signal in the |
| | | | | | WCO2 and SCO2 spectral bands |
| | $\delta\nabla_{CO_2}$ | $\delta\nabla_{CO_2}$ | $\delta\nabla_{CO_2}$ | $\delta\nabla_{CO_2}$ | "CO$_2$ grad del" (see text) |
| | $\log(AOD_{dust})$ | – | – | – | Logarithm of dust AOD |
| | H$_{ice}$ | – | – | – | Vertical height of ice cloud |
| | – | EOF$^3_{SCO2}$ | – | – | Third Empirical Orthogonal Function |
| | | | | | in the SCO$_2$ spectral band |
| | – | – | $dP$ | $dP$ | Retrieved minus a priori surface pressure |
| | – | t$_{year}$ | – | – | Time in years |



**Table 7.** Comparison of ACOS GOSAT v7.3 versus v9 sounding throughput and XCO$_2$ scatter against truth metrics before and after filtering and bias correction.

| Mode | Truth Metric | Version | N (soundings) | Throughput (%) | Sigma (ppm) Unfiltered Raw | QF & BC |
|------|-------------|---------|---------------|----------------|----------------------------|---------|
| Ocean-Glint | Models | v7.3 | 82k | 78% | 1.7 | 1.0 |
| | | v9 | 1131k | 56% | 1.9 | 0.9 |
| | TCCON | v7.3 | 2k | 77% | 2.0 | 1.2 |
| | | v9 | 15k | 58% | 2.4 | 1.1 |
| Land H-gain | Models | v7.3 | 546k | 37% | 5.2 | 1.5 |
| | | v9 | 760k | 37% | 5.1 | 1.4 |
| | TCCON | v7.3 | 5k | 45% | 4.3 | 1.7 |
| | | v9 | 56k | 47% | 4.4 | 1.6 |
| Land M-gain | Models | v9 | 286k | 39% | 2.7 | 1.1 |
| | TCCON | v9 | 9k | 56% | 2.8 | 1.1 |

Figure 2 shows the relative magnitudes of the bias correction on the good quality soundings by season, aggregated to 2.5° latitude by 5° longitude. The global median bias of -1.8 ppm has been removed for clarity. This highlights gradients and contrasts in the bias correction, which are of importance as gradients in CO$_2$ concentrations are the primary driver of CO$_2$ fluxes in atmospheric inversion systems. In general, the bias correction is necessary to remove spurious contrasts between land and ocean-glint XCO$_2$ values. The strongest relative bias corrections are positive adjustments over the bright land surfaces in M-gain viewing mode, specifically the Sahara in DJF and JJA, and Australia in DJF. The Land H-gain observations have a mix of relative bias correction values, ranging from mildly negative over high northern latitudes in JJA, to moderately positive over northern mid-latitudes in JJA in the western U.S. and the Middle East. Most of the Ocean-Glint observations have a mildly negative relative bias correction, with some mild positive values in the southern tropical oceans in DJF.

## 4   Evaluation of ACOS GOSAT v9 XCO$_2$

The ACOS GOSAT v9 XCO$_2$ record was characterized in five ways: (i) an analysis of the XCO$_2$ "good quality" data volume, (ii) a spatiotemporal analysis of the XCO$_2$ estimates, (iii) a validation against XCO$_2$ estimates from TCCON, (iv) a comparison of to XCO$_2$ derived from models, and (v) a comparison with collocated XCO$_2$ estimates from the OCO-2 v10 product.



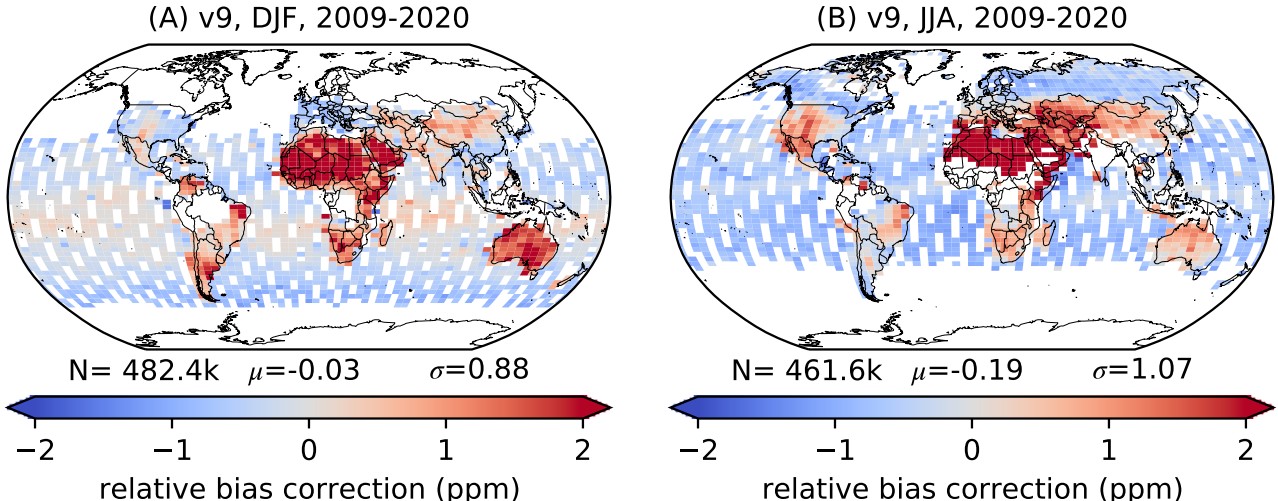

**Figure 2.** Maps of ACOS GOSAT v9 relative bias correction for DJF (A) and JJA (B) for good QF soundings in the eleven year data record at 2.5° by 5° latitude/longitude, after removal of the global, median bias of -1.8 ppm. Grid cells with less than 5 GOSAT soundings are not colored. The total number of soundings (N), and the mean bias ($\mu$) and standard deviation of the bias ($\sigma$) in each grid cell are given.

## 4.1 ACOS GOSAT v9 "good quality" data volume

It is instructive to compare the ACOS GOSAT v9 product to the earlier v7.3 product to highlight similarities and differences in the quality filter screening. A time series histogram of the monthly throughput of the good quality filtered soundings for the v9 product compared to v7.3 is shown in Figure 3. The soundings have been binned by month, with the three GOSAT observation modes displayed by color. The v7.3 product did not contain any Land M-gain data in the L2Lite files (red in the figure) as the quality filtering and bias correction were not developed for that gain mode in v7.3 due to some unreconciled

differences. An important feature of the v9 data record is the extension in time, which runs through June 2020, compared to a termination date of June 2016 for v7.3. Even for the overlapping v7.3 and v9 time period (2009 through mid 2016), there are some differences in the data volume for Land H-gain and Ocean-Glint observations. This is due to changes in both the details of the QF procedure, including changes in the variable thresholds used to assign QF=good/bad, and to some differences in the convergence characteristics of the L2FP retrieval. Generally, v9 is producing up to 60% more good-quality data than v7.3 near

the end of the overlap period in 2016. There was a substantial increase in the number of good QF soundings from 2010 to 2019, due to the increased latitudinal range of the ocean observations as a result of improvements in the GOSAT pointing strategy, as well as improvements in the sounding selection for ACOS L2FP v9.

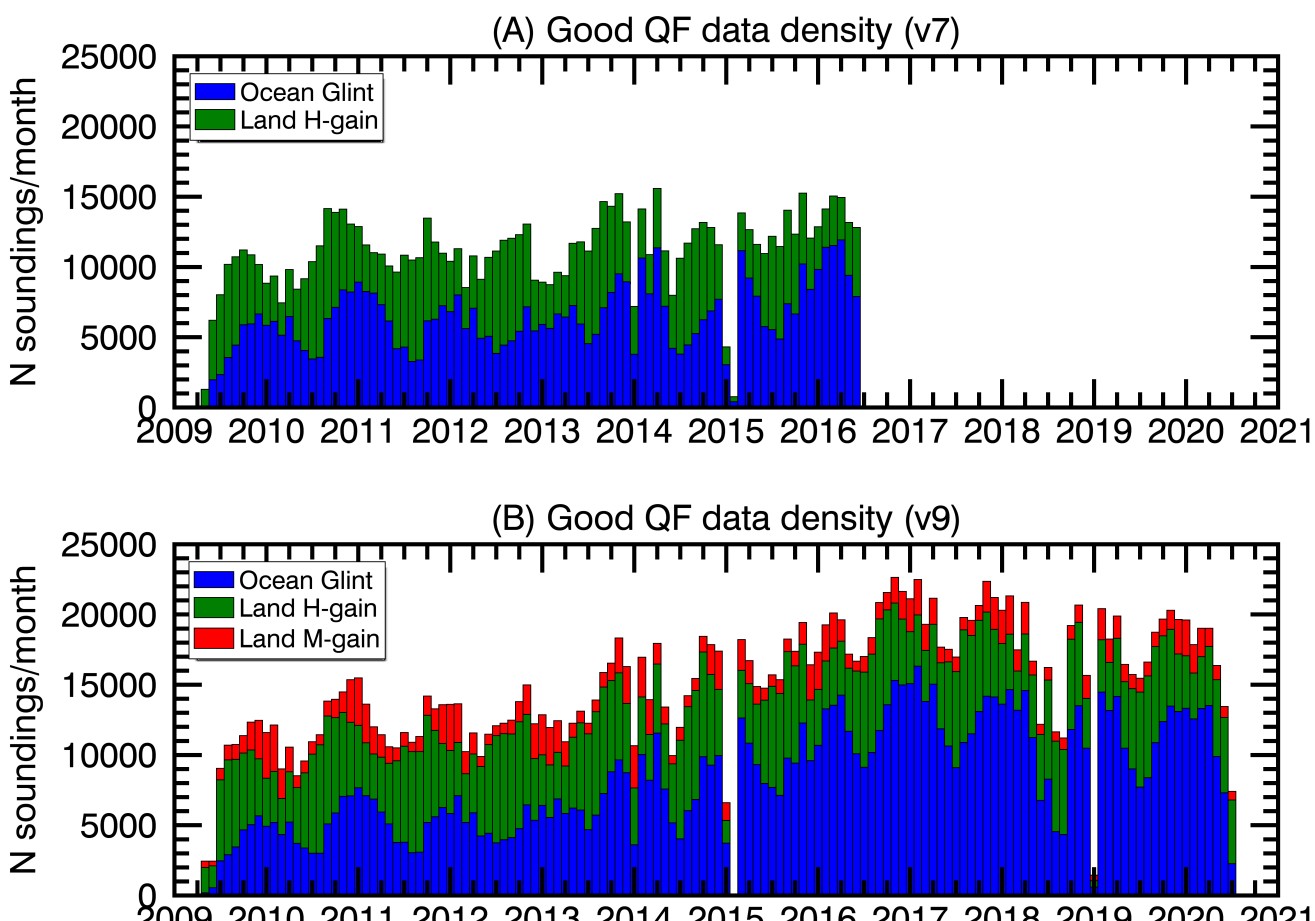

**Figure 3.** Time series histograms of the monthly number of good QF GOSAT soundings for v7.3 (top) and v9 (bottom) spanning the eleven year record. The large data gaps in early 2015 and 2019 were caused by a switch from the primary to secondary pointing mirror and a solar panel failure, respectively.

Figure 4 shows sounding density Hovmöller plots comparing ACOS GOSAT v7.3 (A) to v9 (B) with the three GOSAT obser-
vation modes combined. Again, the extended time period covered by v9 is evident. The increase in sounding density in the SH
beginning in 2016 due to optimization of the GOSAT viewing strategy is prominent in the v9 product. This feature is also seen
in the spatial maps showing the fraction of good quality soundings and the density per grid box, in panels C and D of Figure 1,
which was introduced in Section 3.2. Persistently clear regions, such as the Sahara and western Australia, have as many as
30% of the observations assigned a good quality flag. Large regions of the tropical Pacific and Atlantic also contain a relatively
high fraction of good quality soundings. On the other hand, tropical forests and high latitudes in general have low yields of
good quality soundings. This is largely a combination of cloud contamination, dark surfaces at shortwave infrared wavelengths,
and low solar illumination conditions, all three of which are problematic for retrieving $CO_2$ from space using reflected sunlight.

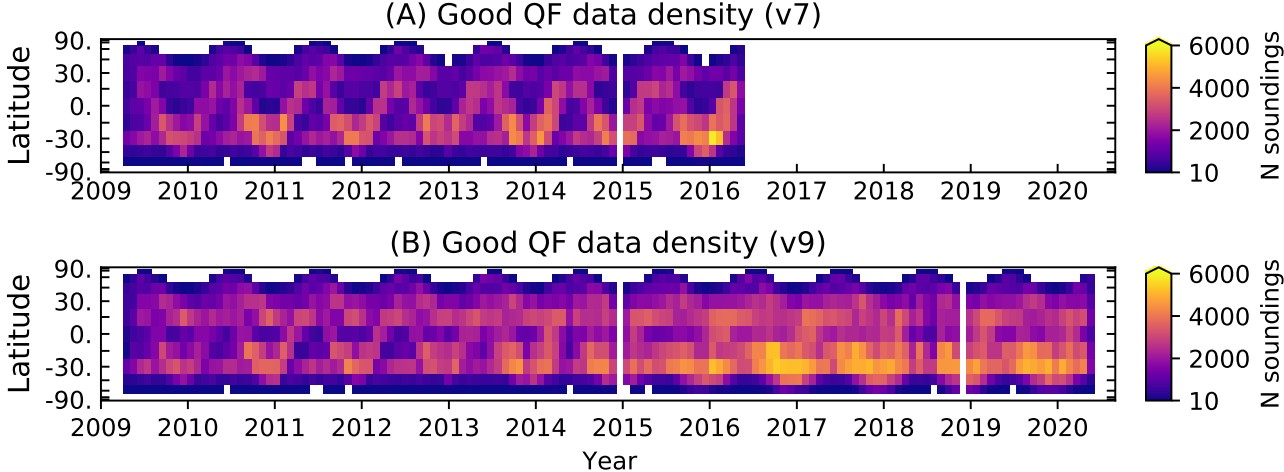

**Figure 4.** Sounding density comparing v7.3 (A) to v9 (B) good QF data as a function of time and latitude at 30 day by 15° latitude resolution
for all viewing modes combined.

## 4.2  ACOS GOSAT v9 XCO$_2$ spatiotemporal analysis

There has been a steady increase in the atmospheric burden of $CO_2$ since the onset of the industrial age due mainly to the
burning of fossil fuels (e.g., Keeling et al., 1995). In May of 2009, at the beginning of the GOSAT mission, the mean global
value of XCO$_2$ reported by the NOAA Global Monitoring Laboratory was 387.95 ppm, while by May of 2020, the mean global
value had risen to 413.81 ppm (Dlugokencky and Tans, 2021). This yields a secular increase of $\simeq 2.35$ ppm/yr. For compari-
son, Figure 5 shows the ACOS GOSAT v9 bias corrected and quality filtered XCO$_2$ as a function of latitude (15° increments)
and time (30 d increments) for Ocean-Glint observations (A), and combined Land M and H-gain observations (B). Using the
monthly mean XCO$_2$ values (combined land and ocean) for May 2009 (386.50 ppm) and May 2020 (411.82 ppm), the ACOS
GOSAT v9 record has a secular increase of $\simeq 2.30$ ppm/yr over the eleven year record. This small disagreement in secular trend

of approximately 2% is understandable, given the significant differences in the spatiotemporal sampling of the two data sets.
For the interested reader, a thorough comparison of satellite and surface-derived growth rates in atmospheric $CO_2$ is given in
Buchwitz et al. (2018).

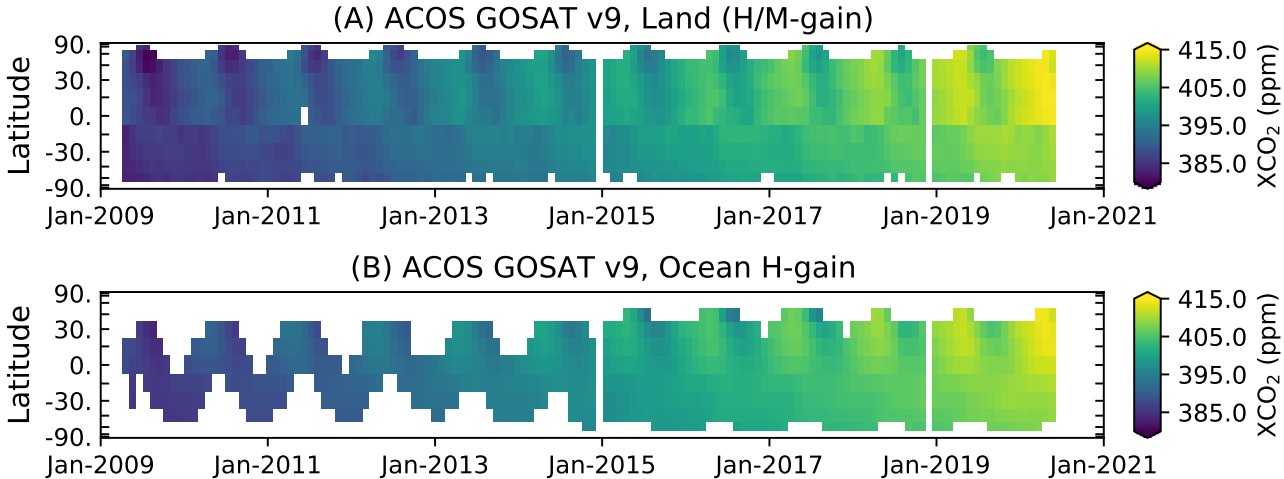

**Figure 5.** ACOS GOSAT v9 bias corrected and quality filtered $XCO_2$ as a function of latitude (15° increments) and time (30 d increments) for Ocean-Glint observations (A), and combined Land M-gain and H-gain observations (B). Grid cells with less than 10 GOSAT soundings are not colored.

The maps in Figure 6 show the spatial distribution of $XCO_2$ at 2.5° latitude by 5° longitude resolution for 2010 (top) and 2019 (bottom), for DJF (left) and JJA (right). The dynamic range of the color scale in each case is 6 ppm. However, due to the secular increase in global $CO_2$ of $\simeq 2.3$ ppm per year, the scale is centered $\simeq 20$ ppm higher in 2019 compared to 2010. The strong
latitudinal gradients in $XCO_2$ are similar in these two seasons, while the zonal gradient tends to be weakest in MAM (not shown), just before the summer draw down of $CO_2$ by the land biosphere begins. The increase in the number of Ocean-Glint soundings in the later part of the data record is also evident in these maps.

Qualitatively, the patterns in the maps look quite similar from 2010 to 2019, but with increased data coverage. In general, the
highest concentrations of $XCO_2$ for the two selected seasons are observed by GOSAT in the northern hemisphere (NH) during DJF, especially over northern tropical Africa (between 0° and 15°N latitude), large portions of China, and the eastern United States. This stands to reason, as the atmospheric burden of $CO_2$ increases towards a peak during NH winter due to inactivity of the land biosphere, coupled with strong anthropogenic $CO_2$ emissions. During DJF the ACOS GOSAT v9 $XCO_2$ exhibits relatively low concentrations across the entire SH, as would be expected if the Southern ocean were a strong carbon sink (e.g.,



Gruber et al., 2019)). In JJA, the XCO$_2$ is reduced over the mid-latitude and boreal forests, also expected behavior due to strong
photosynthetic uptake of CO$_2$ during this season (e.g., Ciais et al., 2019).

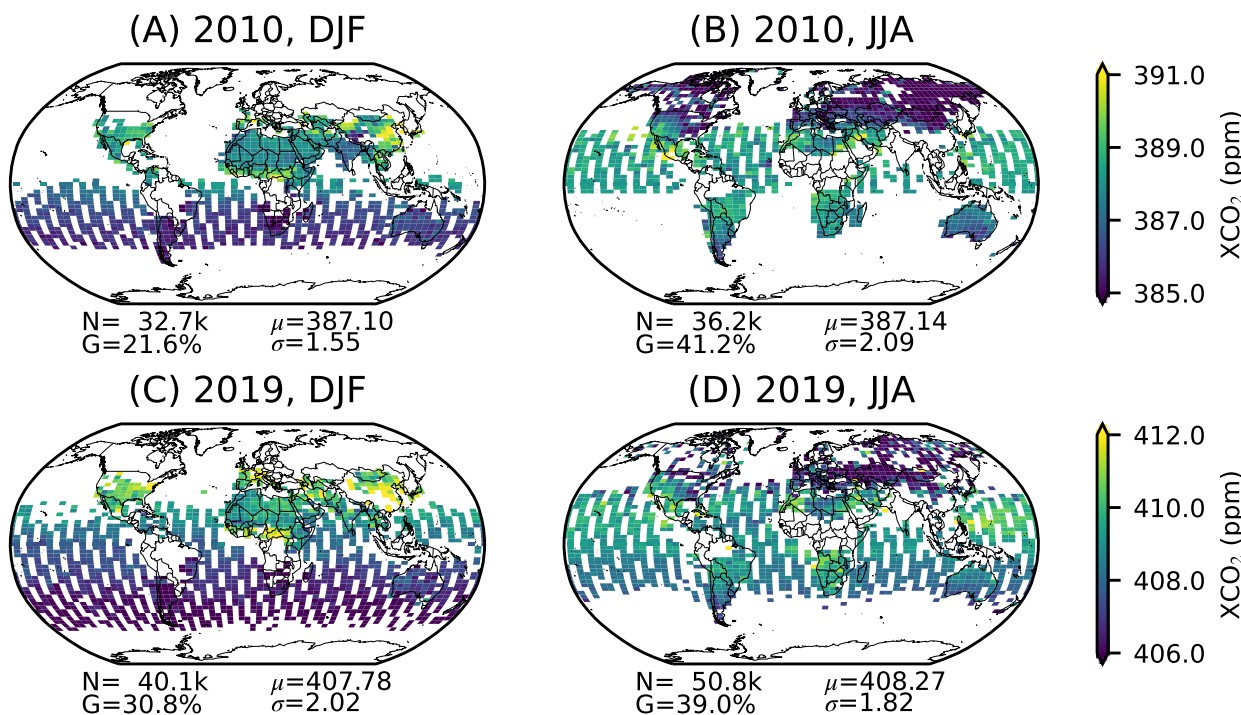

**Figure 6.** ACOS GOSAT v9 bias corrected XCO$_2$ for the good QF soundings at 2.5° latitude by 5° longitude resolution for DJF 2009-2010
(A), JJA 2010 (B), DJF 2018-2019 (C), and JJA 2019 (D). The dynamic range of the color scale in each case is 6 ppm. However, due to the
secular increase in global CO$_2$ of $\simeq$ 2.2 ppm per year, the scale is centered $\simeq$ 20 ppm higher in 2019 compared to 2010. Grid cells with less
than 5 GOSAT soundings are not colored.

### 4.3 ACOS GOSAT v9 XCO$_2$ versus TCCON

A list of TCCON stations used in this work, including basic physical information and data citations, is given in Table 8. For the
evaluation against the ACOS v9 XCO$_2$ data, the single sounding collocations described in Section 3.3, were aggregated into
overpass mean values. Essentially the same TCCON dataset was used for both the QF/BC procedure as for the evaluation, as
no hold-over data was maintained. Also, as described in Section 3.3, an averaging kernel correction was applied to the TCCON
data in order to fairly compare to the satellite data. A one-to-one linear regression of the XCO$_2$ provides a simple quantification
of the agreement, as shown in Figure 7.





**Table 8.** List of TCCON stations used in the BC/QF and XCO$_2$ evaluation of ACOS GOSAT v9, along with the data citations.

| TCCON Station Name | Continent *=island | Latitude (degrees) | Altitude (meters) | Operational Date Range (YYYYMM - YYYYMM) | Data Citation |
|---|---|---|---|---|---|
| Eureka | North America | 80.1 N | 610 | 200608 – present | Strong et al. (2019) |
| Sodankylä | Europe | 67.4 N | 188 | 200901 – present | Kivi et al. (2014) |
| East Trout Lake | North America | 54.4 N | 502 | 201610 – present | Wunch et al. (2017a) |
| Bialystok | Europe | 53.2 N | 180 | 200903 – 201810 | Deutscher et al. (2019) |
| Bremen | Europe | 53.1 N | 27 | 200407 – present | Notholt et al. (2019) |
| Karlsruhe | Europe | 49.1 N | 116 | 200909 – present | Hase et al. (2015) |
| Paris | Europe | 48.8 N | 60 | 201409 – present | Te et al. (2014) |
| Orleans | Europe | 48.0 N | 130 | 200908 – present | Warneke et al. (2019) |
| Garmisch | Europe | 47.5 N | 740 | 200707 – present | Sussmann and Rettinger (2018a) |
| Zugspitze | Europe | 47.4 N | 2960 | 201204 – present | Sussmann and Rettinger (2018b) |
| Park Falls | North America | 45.9 N | 440 | 200405 – present | Wennberg et al. (2017) |
| Rikubetsu | Asia | 43.5 N | 380 | 201311 – present | Morino et al. (2016) |
| Indianapolis | North America | 39.9 N | 270 | 201208 – 201212 | Iraci et al. (2016b) |
| Lamont | North America | 36.6 N | 320 | 200807 – present | Wennberg et al. (2016b) |
| Four Corners | North America | 36.8 N | 1643 | 201103 – 201310 | Dubey et al. (2014); Lindenmaier et al. (2014) |
| Anmyeondo | Asia | 36.5 N | 30 | 201408 – present | Goo et al. (2014) |
| Tsukuba | Asia* | 36.1 N | 30 | 200812 – present | Morino et al. (2018a) |
| Nicosia | Europe* | 35.1 N | 185 | 201908 – present | Petri et al. (2020) |
| Edwards | North America | 35.0 N | 699 | 201307 – present | Iraci et al. (2016a) |
| JPL | North America | 34.2 N | 390 | 201103 – 201307 | Wennberg et al. (2016a) |
| | | | | 201706 – 201805 | Wennberg et al. (2016a) |
| Caltech | North America | 34.1 N | 230 | 201209 – present | Wennberg et al. (2015) |
| Saga | Asia* | 33.2 N | 7 | 201106 – present | Kawakami et al. (2014) |
| Hefei | Asia | 31.9 N | 29 | 201509 – 201612 | Liu et al. (2018) |
| Izana | Africa* | 28.3 N | 237 | 200705 – present | Blumenstock et al. (2017) |
| Burgos | Asia* | 18.5 N | 35 | 201703 – present | Morino et al. (2018b) |
| Ascension | Africa* | 7.9 S | 10 | 201205 – present | Feist et al. (2014) |
| Darwin | Australia | 12.4 S | 30 | 200508 – present | Griffith et al. (2014a) |
| Reunion | Africa* | 20.9 S | 87 | 201109 – present | De Mazière et al. (2017) |
| Wollongong | Australia | 34.4 S | 30 | 200805 – present | Griffith et al. (2014b) |
| Lauder | Australia* | 45.0 S | 370 | 200406 – present | Pollard et al. (2019) |



**Figure 7.** Quality filtered and bias corrected ACOS GOSAT v9 vs collocated TCCON XCO$_2$ for Ocean-Glint (A), Land H-gain (B), and Land M-gain (C). Each point represents an overpass mean, which typically contain 5-10 GOSAT soundings per overpass. The legend in the lower right indicates the number of collocated overpass means for individual TCCON stations. Summary statistics for all stations combined are reported in the upper left of each panel for both single sounding and overpass means.





For Ocean-Glint observations (A), the mean ($\mu$) of the differences in $XCO_2$ ($\Delta XCO_2^{TCCON}$ = GOSAT - TCCON) is essentially zero: 0.00 ppm for the single-sounding (SS) results, and +0.01 ppm for the overpass mean (OPM) results. The corresponding standard deviations ($\sigma$) are 1.08 ppm and 0.82 ppm for the SS and OPM results, respectively. This indicates that roughly half of the SS error variance is a result of instrument noise, or other random high frequency error sources ($1.08^2$=1.2 ppm versus $0.82^2$=0.7 ppm).


  For Land H-gain observations (B), $\mu$=+0.10 ppm and +0.14 ppm for the SS and OPM, respectively. The Land H-gain $\sigma$ are higher than for Ocean-Glint; 1.60 and 1.14 ppm for SS and OPM, respectively. Larger variations in $\Delta XCO_2^{TCCON}$ are expected for Land H-gain due to variability in topography and surface brightness, as well as higher likelihood of contamination by cloud and aerosol, all of which are more challenging for the ACOS retrieval. Further, biology and atmospheric transport cause $CO_2$

signals to vary more over land regions, and in addition, instrument noise is higher because the SNRs tend to be lower.

  Land M-gain observations have near zero bias ($\mu$=-0.02 ppm and +0.02 ppm for SS and OPM, respectively), and scatter similar to that for Ocean-Glint ($\sigma$=1.09 and 0.84 ppm for SS and OPM, respectively), likely driven by lower variability in surface topography and brightness compared to Land H-gain observations, as well as higher SNRs over these bright land surfaces.


  The correlation in the $XCO_2$ between the datasets in all observation modes is high, with Pearson $R^2$=0.98, 0.98, and 0.99 for Ocean-Glint, Land H-gain, and Land M-gain, respectively. Overall, these results indicate excellent agreement between the bias corrected and quality filtered ACOS GOSAT v9 $XCO_2$ product and collocated estimates from TCCON.

Figure 8 shows the mean absolute error (MAE) between the overpass mean collocated GOSAT and TCCON $XCO_2$, organized by latitude bins, season, and observation mode for v7.3 (Panels A and B) and v9 (Panels C, D, E). The error bars on the MAE represent the scatter around the mean. A smaller error bar, or a lower scatter, implies that the MAE values are more consistent across a group of TCCON stations within a latitude band and season. The calculation of the MAE and error bars follow the procedure reported in Chatterjee et al. (2013) (Equations 3 and 4). The MAEs tend to be lower for v9 compared to v7.3, with

smaller error bars, and increased number of collocations. This is especially true for the SH Ocean-Glint data, where the MAE ranges from 0.4 to 0.7 ppm in v9 for all seasons, in contrast to v7.3, which had higher MAE ranging from 0.5 to 0.85 ppm in that region. In the v9 Land H-gain data, the MAE is roughly a function of latitude, with the highest values ($\simeq 1.0$ ppm) seen between 60°N – 90°N, and the lowest values ($\simeq 0.7$ ppm) seen from 30°S – 60°S. The error bars on the v9 Land H-gain estimates in the 30°N – 60°N latitude range are very small, due in part to the large number of collocations. There is very limited

land data between 0° and 30°N (approximately 25% of Earth's surface), due to the sparsity of TCCON stations in this latitude band. Only Burgos (18.5°N) and Izana (28.3°N) are located in this range (reference Table 8), and many of the collocations from these sites are from GOSAT observations made in Ocean-Glint viewing.

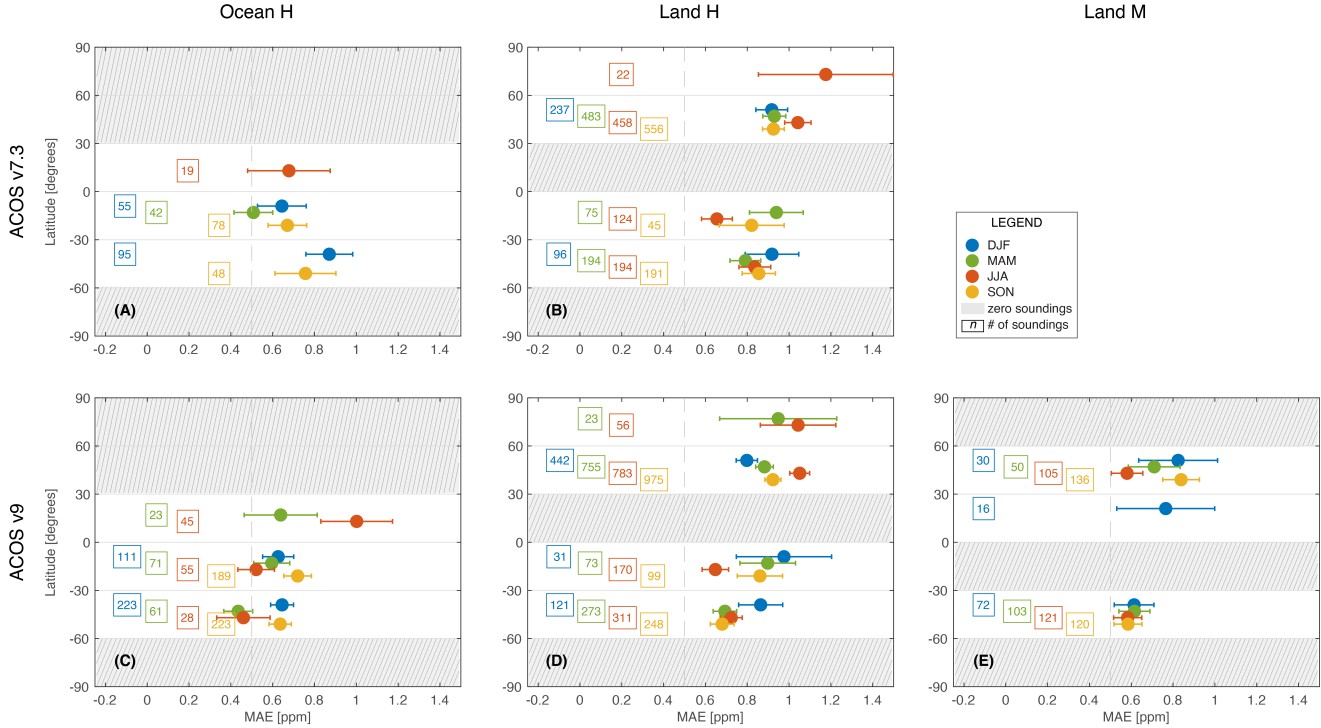

**Figure 8.** Mean Absolute Error (MAE; ppm) between ACOS GOSAT v9 XCO$_2$ and collocated TCCON observations for v7.3 (panels A and B) and v9 (panels C, D and E), binned by latitude for: Ocean-Glint (panels A and C), Land H-gain (panels B and D) and Land M-gain (panel E). Recall that there was no Land M-gain data generated for v7.3. The error bars on the MAE represent the standard error of the mean. The calculation of the MAE statistic and error bars follow the procedure reported in Chatterjee et al. (2013) (Equations 3 and 4). The total number of independent observations available within each latitude band and for each season are reported in the colored boxes. Latitude bands that do not have any collocated soundings are shaded in grey.

Knowledge of the average XCO$_2$ seasonal cycle can be used to disentangle the CO$_2$ growth rate from the seasonal variability,
as well as for quantifying potential seasonal biases between satellite and ground-based XCO$_2$ estimates. Lindqvist et al. (2015) fitted a skewed sine wave (See Eq. 1 of Lindqvist et al., 2015) to the ACOS GOSAT v3.5 XCO$_2$ time series and the TCCON estimates of XCO$_2$ at 16 stations, spanning April 2009 through December 2013. They found that ACOS GOSAT v7.3 captured the seasonal cycle within approximately 1 ppm of the TCCON estimates for all but the European sites, and that the satellite and ground-based CO$_2$ growth rates agreed generally better than 0.2 ppm per year. Here, we provide an update to those results using
the eleven year ACOS GOSAT v9 XCO$_2$ data record. For this part of the analysis, a slightly more restrictive set of collocation criteria were implemented, compared to that described in Section 3.3 for the BC/QF procedure and to that used to generate Fig. 7. The seasonal cycle analysis required that the TCCON record spanned at least one contiguous year (a full seasonal cycle), and that a minimum of 20 collocations with GOSAT occurred. In addition, the three GOSAT observation modes (Ocean-Glint,





Land M-gain, Land H-gain) were combined for each site, and satellite overpass means of $XCO_2$ were aggregated into daily
means. This resulted in approximately 7700 daily averages at 26 TCCON stations over the 11 year GOSAT data record.

Figure 9 shows the results of the seasonal cycle fit for the Lamont, Oklahoma TCCON station. The one-to-one scatter of the
896 daily averaged $XCO_2$ values (A) indicates a bias of -0.27 ppm for the GOSAT product relative to TCCON, with a standard
deviation of 1.25 ppm and a Pearson's $R^2$ of 0.99. The seasonal cycle fits (B) indicate excellent agreement in the secular $CO_2$
increase at this site; 2.34 ppm for both GOSAT and TCCON. The mean seasonal amplitudes indicate a slight disagreement
of a few tenths of a ppm, with TCCON showing a slightly higher fitted peak $XCO_2$ value during the spring maximum phase,
compared to GOSAT. This is similar to the results for this site reported in (Fig. 4 of Lindqvist et al., 2015). The time series
of the calculated difference in satellite and ground-based estimated $XCO_2$ (GOSAT - TCCON), shown in (C), highlights the
magnitude of the scatter about the mean bias, and suggests that there is no observable time-drift in the data at this site.


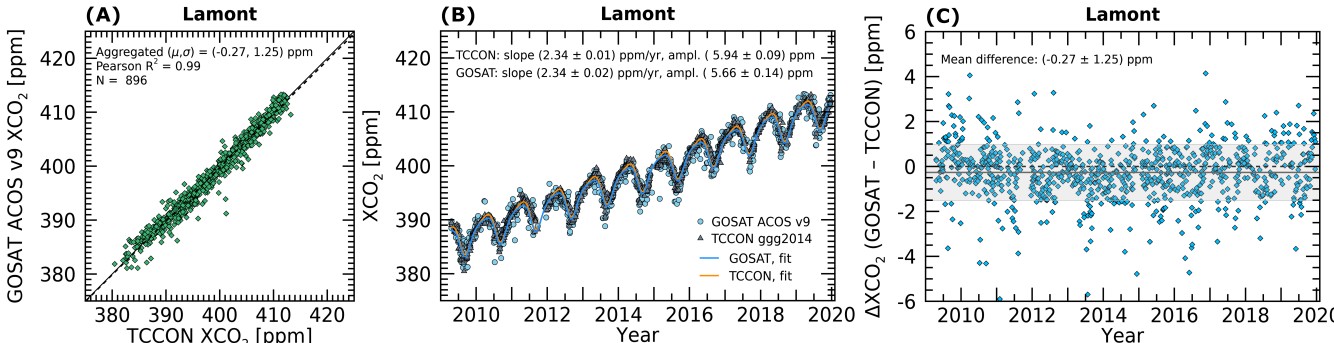

**Figure 9.** Seasonal cycle analysis of ACOS GOSAT v9 $XCO_2$ versus collocated TCCON for the Lamont, Oklahoma station, following the
methodology of Lindqvist et al. (2015). Panel (A) shows the one-to-one scatter plot of the bias corrected daily mean $XCO_2$ for GOSAT
versus TCCON. Panel (B) shows the time series of GOSAT $XCO_2$ (blue circles) with fit (blue line) and the TCCON $XCO_2$ (grey triangles)
with fit (orange line) over the eleven year data record. Panel (C) shows the time series of calculated $\Delta XCO_2^{TCCON}$ (GOSAT - TCCON), with
the mean difference (horizontal solid black line) and $\pm 1$ standard deviation (gray shading). The three GOSAT observation modes have been
combined into daily mean averages to provide the maximum number of collocations possible for the seasonal fits.

A summary of the data from each station that met the seasonal cycle collocation criteria is provided in Table 9. In addition,
the full complement of plots are presented in Appendix A. Overall, the seasonal cycle analysis at most sites is in agreement, to
within the estimated uncertainties. The standard deviation of the mean $XCO_2$ biases for the 26 sites is 0.41 ppm for the ACOS
GOSAT v9 record. This compares to a value of 0.51 ppm at 23 stations for ACOS GOSAT v7.3, suggesting an improvement in
the quality of the v9 $XCO_2$ product.



**Table 9.** Evaluation of the daily mean bias corrected ACOS GOSAT v9 XCO₂ (all viewing modes combined) against collocated TCCON estimates for individual stations. There were 7547 days total for the 25 stations. The following sites/instruments were excluded from this part of the analysis due to inadequate timeseries or seasonal cycle coverage; Eureka, Four Corners, Indianapolis (Influx), JPL2007, Lauder1, Lauder3, Manaus, and Ny Ålesund. The mean, standard deviation, and Pearson correlation coefficient ($\mu$, $\sigma$, $R^2$) of the linear fit between GOSAT and TCCON are given in columns 3-5. The remaining columns quantify the seasonal cycle fit following the methodology described in Lindqvist et al. (2015). The bottom row provides mean summary statistics for the linear fit.

| TCCON Station Name | N (days) | $\Delta$XCO₂ (GOSAT - TCCON) $\mu$ (ppm) | $\sigma$ (ppm) | Linear Correlation ($R^2$) | Trend±uncertainty (ppm/yr) TCCON | GOSAT | Amplitude±uncertainty (ppm) TCCON | GOSAT |
|---|---|---|---|---|---|---|---|---|
| Sodankylä | 166 | 0.91 | 1.57 | 0.98 | 2.32 ±0.02 | 2.37 ±0.04 | 9.83 ±0.95 | 10.30 ±2.76 |
| East Trout Lake | 86 | 0.69 | 1.52 | 0.94 | 2.19 ±0.11 | 1.98 ±0.23 | 10.30 ±0.24 | 9.29 ±0.54 |
| Bialystok | 252 | 0.01 | 1.26 | 0.98 | 2.29 ±0.02 | 2.38 ±0.04 | 8.97 ±0.13 | 8.67 ±0.26 |
| Bremen | 95 | 0.32 | 2.01 | 0.96 | 2.28 ±0.04 | 2.42 ±0.08 | 8.14 ±0.27 | 8.48 ±1.79 |
| Karlsruhe | 328 | 0.61 | 1.60 | 0.98 | 2.34 ±0.02 | 2.43 ±0.03 | 7.72 ±0.14 | 7.19 ±0.39 |
| Paris | 137 | -0.28 | 1.95 | 0.92 | 2.10 ±0.07 | 2.65 ±0.10 | 7.89 ±0.31 | 6.68 ±0.42 |
| Órleans | 300 | 0.26 | 1.57 | 0.98 | 2.32 ±0.02 | 2.41 ±0.03 | 7.87 ±0.13 | 6.96 ±0.26 |
| Garmisch | 269 | 0.57 | 1.66 | 0.98 | 2.30 ±0.02 | 2.28 ±0.03 | 7.64 ±0.16 | 7.52 ±0.35 |
| Zugspitze | 107 | -0.09 | 2.10 | 0.92 | 2.52 ±0.04 | 2.66 ±0.12 | 6.38 ±0.23 | 8.27 ±0.65 |
| Park Falls | 389 | -0.12 | 1.35 | 0.98 | 2.32 ±0.01 | 2.30 ±0.03 | 8.94 ±0.14 | 9.27 ±0.27 |
| Rikubetsu | 58 | -0.34 | 1.48 | 0.98 | 2.45 ±0.07 | 2.85 ±0.13 | 10.62 ±0.41 | 12.19 ±0.75 |
| Lamont | 896 | -0.27 | 1.25 | 0.99 | 2.34 ±0.01 | 2.34 ±0.02 | 5.94 ±0.09 | 5.66 ±0.14 |
| Anmyeondo | 24 | 0.72 | 1.62 | 0.94 | 2.64 ± 0.22 | 2.93 ± 0.27 | 8.78 ± 0.68 | 10.25 ± 1.06 |
| Tsukuba | 389 | 0.75 | 1.71 | 0.97 | 2.54 ±0.03 | 2.43 ±0.04 | 6.84 ±0.20 | 7.44 ±0.30 |
| Edwards | 543 | 0.38 | 0.91 | 0.98 | 2.45 ±0.01 | 2.46 ±0.02 | 5.51 ±0.08 | 5.63 ±0.14 |
| JPL | 361 | -0.12 | 1.31 | 0.98 | 2.44 ±0.02 | 2.43 ±0.03 | 5.30 ±0.12 | 6.12 ±0.22 |
| Caltech | 852 | 0.71 | 1.26 | 0.97 | 2.44 ±0.02 | 2.49 ±0.02 | 5.74 ±0.10 | 5.85 ±0.14 |
| Saga | 281 | -0.03 | 1.50 | 0.97 | 2.31 ±0.02 | 2.39 ±0.04 | 6.59 ±0.14 | 7.21 ±0.27 |
| Hefei | 38 | -0.22 | 1.77 | 0.78 | 3.22 ±0.49 | 2.79 ±0.72 | 6.60 ±0.87 | 5.64 ±0.88 |
| Izana | 180 | -0.08 | 1.03 | 0.99 | 2.40 ±0.01 | 2.26 ±0.02 | 5.70 ±0.11 | 5.76 ±0.18 |
| Burgos | 80 | -0.43 | 1.09 | 0.91 | 2.21 ±0.07 | 2.38 ±0.16 | 5.86 ±0.23 | 5.31 ±0.46 |
| Ascension | 310 | 0.49 | 0.73 | 0.98 | 2.37 ± 0.01 | 2.30 ±0.00 | 0.32 ±0.10 | 0.62 ±0.10 |
| Darwin | 565 | 0.04 | 1.20 | 0.98 | 2.39 ±0.01 | 2.32 ±0.02 | 0.44 ±0.09 | 3.20 ±0.20 |
| Reunion | 309 | 0.11 | 0.84 | 0.99 | 2.38 ±0.02 | 2.39 ±0.02 | 1.57 ±0.15 | 1.25 ±0.13 |
| Wollongong | 532 | -0.17 | 1.11 | 0.99 | 2.39 ±0.01 | 2.36 ±0.02 | 1.03 ±0.09 | 1.34 ±0.18 |
| Lauder 2 | 194 | -0.31 | 1.64 | 0.96 | 2.35 ±0.01 | 2.23 ±0.02 | 0.39 ±0.10 | 0.30 ±0.36 |
| Mean | 302 | 0.18 | 1.42 | 0.96 | | | | |



## 4.4 ACOS GOSAT v9 XCO$_2$ versus models

The collocation and calculation of the multi-model-mean (MMM) was described in Section 3.3. Although the model data used for evaluation was very similar to that used in the QF/BC procedure, some minor version updates and extensions in time were

included, as indicated in Table 4. It is important to be aware that there can be a considerable time delay between performing the QF/BC procedure and the full generation of the final product, during which time the models are often updated.

Seasonal maps of $\Delta XCO_2^{MMM}$ (GOSAT v9 minus MMM) are shown in Figure 10 for the eleven year data record binned at 2.5° latitude by 5° longitude. Generally, the agreement between the model derived values and the satellite estimates is quite good,

with an annual mean difference of $\simeq$ -0.15 ppm, and binned scatter $\simeq$ 0.5 ppm. For Ocean-Glint observations, the $\Delta XCO_2^{MMM}$ tends to be negative (positive) in the northern (southern) hemispheres. Land observations exhibit several distinct sub-continental scale disagreements, including a strong positive signal over northern tropical Africa in DJF (GOSAT XCO$_2$ higher than MMM).

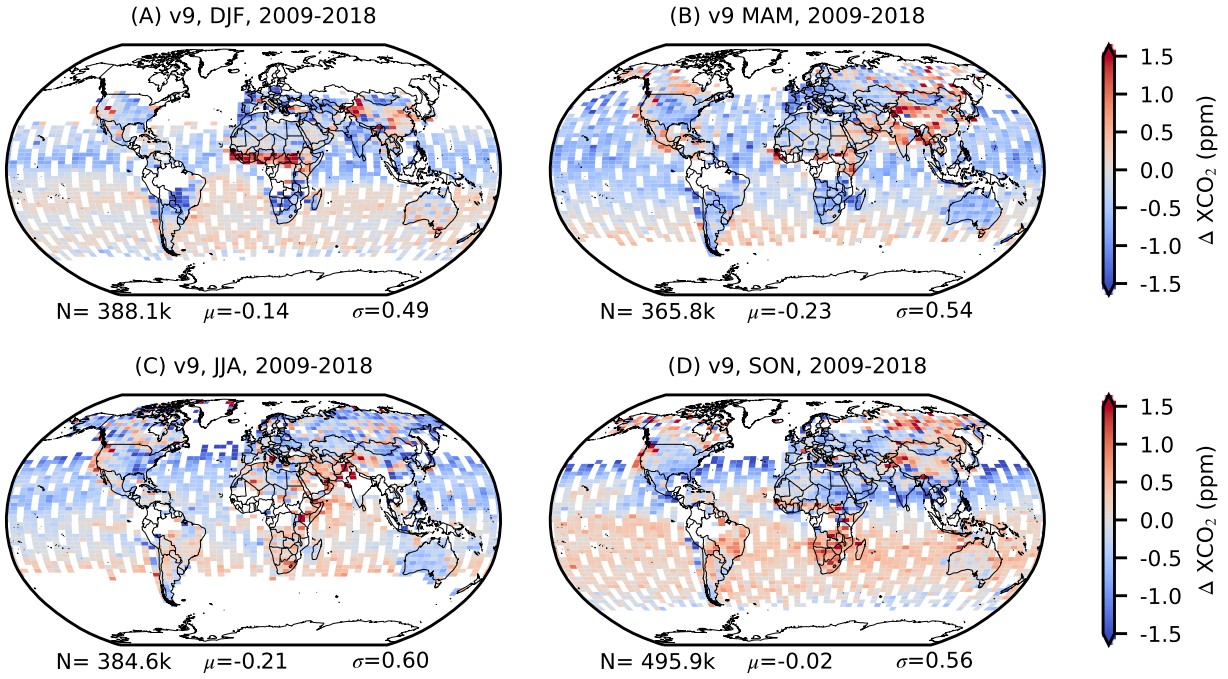

**Figure 10.** Seasonal maps of the mean $\Delta XCO_2^{MMM}$ (GOSAT - MMM) spanning 2009 through 2018 for DJF (A), MAM (B), JJA (C), and SON (D) at 2.5° latitude by 5° longitude resolution. Grid boxes containing less than 10 collocations are colored white.

Figure 11 shows Hovmöller plots of the $\Delta XCO_2^{MMM}$ Ocean-Glint data at 30 day by 15° latitude resolution for v7.3 (A) and v9

(B). The extension in time of v9 is evident, as well as the expansion in the latitude range of the Ocean-Glint observations since

2015. A direct comparison between the v7.3 and v9 $\Delta XCO_2^{MMM}$ values for the overlapping time period, April 2009 through June 2016, reveals a global mean bias and standard deviation of -0.54 ppm and 1.0 ppm for the v7.3 product, and -0.20 ppm and 0.84 ppm for v9, underscoring the improvement.

Of particular note are the strong positive $\Delta XCO_2^{MMM}$ values in the v9 SH Ocean-Glint observations for the latter part of 2014, persisting through most of 2015. This feature is is not seen in the v7.3 product, due to a paucity of SH Ocean-Glint data. It approximately coincides with the strong 2015-2016 El Ninõ event, where $\Delta XCO_2^{MMM}$ signals were also seen in the OCO-2 v7 Ocean-Glint data, as reported in Chatterjee et al. (2017). It has been hypothesized that the 2015-2016 El Ninõ produced an anomalously strong carbon release from tropical land regions due to higher temperature and below average precipitation (Liu

et al., 2017). In contrast to the positive SH signal, negative $\Delta XCO_2^{MMM}$ values are observed in the v9 NH oceans since 2016. It is unclear why the satellite and models disagree over such large spatial and temporal scales. Further investigation is required in order to assess whether they are artifacts from the Ocean-Glint retrievals, such as due to the specification of the L2FP $CO_2$ prior, or whether it is a process-based response of the carbon cycle. In the future, independent $XCO_2$ evaluation datasets such as that described by Müller et al. (2021) have potential to help resolve such issues.


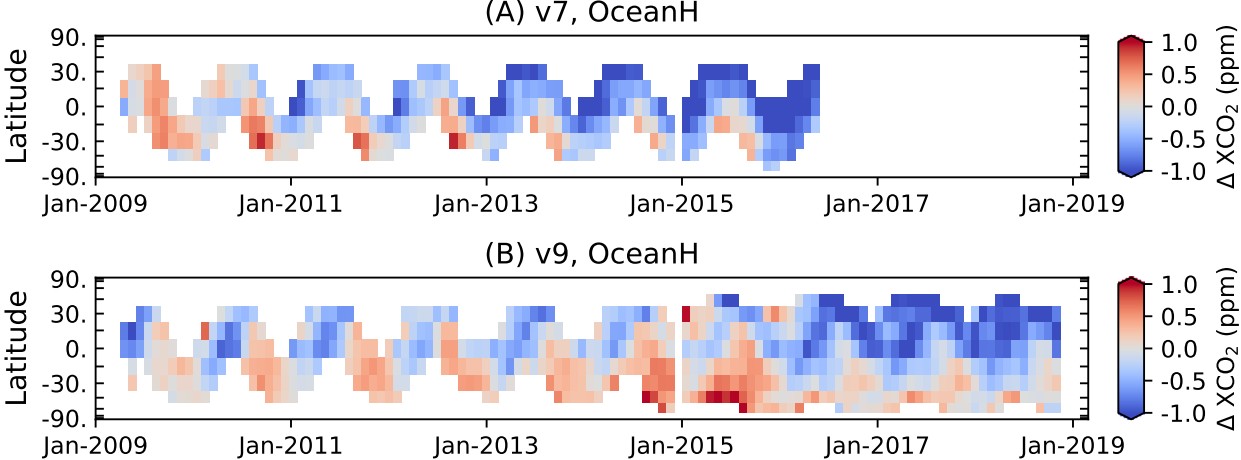

**Figure 11.** Time series of $\Delta XCO_2^{MMM}$ (ACOS GOSAT v9 - MMM) versus latitude at 30 day by 15° resolution for Ocean-Glint observations for v7.3 (A) and v9 (B). Grid cells containing less than 10 collocations are colored white.

Figure 12 shows spatial maps of $\Delta XCO_2^{MMM}$ for the truncated time span 2010 through 2015 comparing v7.3 (A and C) and v9 (B and D) for DJF (A and B) and JJA (C and D). There is significant decrease in scatter in the v9 product ($\simeq 0.5$ ppm) relative to v7.3 ($\simeq 0.75$ ppm). The low bias in DJF tropical Pacific vanishes in v9 and the positive bias in JJA extra-tropical land regions is reduced. The expanded latitudinal extent of the Ocean-Glint observations is evident in the v9 maps. One feature that

is robust in both v7.3 and v9 is the large positive signal over northern tropical Africa in DJF. This feature was also observed in

the OCO-2 v7 and v8 comparisons to a MMM (O'Dell et al., 2018), and in v10 $XCO_2$ anomaly maps (Hakkarainen et al., 2019).

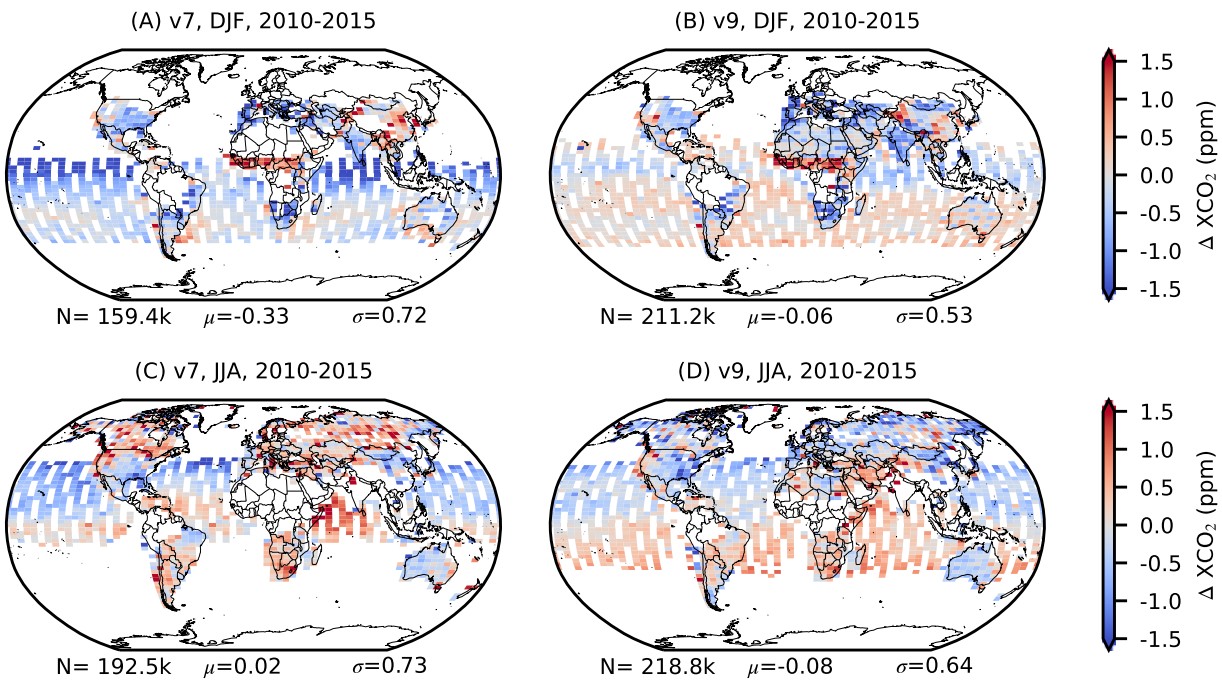

**Figure 12.** Maps of the mean $\Delta XCO_2^{MMM}$ (GOSAT - MMM) spanning 2010 through 2015 for v7.3 DJF (A), v9 DJF (B), v7.3 JJA (C), and v9 JJA (D) at $2.5°$ by $5°$ latitude/longitude resolution. Only grid boxes with at least 10 collocations are shown.

## 4.5 ACOS GOSAT v9 $XCO_2$ versus OCO-2

NASA's Orbiting Carbon Observatory-2 (OCO-2) has been collecting science data since September, 2014 from a near-polar
low-Earth orbit (705 km altitude), with an afternoon equator crossing time of $\simeq 1{:}30\,PM$ (Crisp et al., 2017). Like GOSAT, OCO-2 takes measurements of reflected solar radiation in the Oxygen A-band ($0.76\,\mu m$), and the weak and strong carbon dioxide bands (1.6 and 2.0 $\mu m$, respectively), which are used to estimate $XCO_2$ using the ACOS L2FP retrieval algorithm (Eldering et al., 2017; O'Dell et al., 2018). However, due to differences in the orbit parameters of the two sensors, e.g., a 3 day repeat cycle for GOSAT versus a 16 day repeat cycle for OCO-2 (see Table 2 of Kataoka et al., 2017), the number of collocated
soundings is somewhat limited. Therefore, some criteria must be defined in order to identify soundings that can be compared in a meaningful way. The underlying assumption of the collocation is that on scales of a few hundred kilometers and several hours, the natural variance in $XCO_2$ is not detectable in satellite derived estimates from the ACOS L2FP algorithm.





For this study, the coincidence criteria to match OCO-2 soundings to individual GOSAT soundings were those: (i) falling
within $2°$ latitude and $3°$ longitude, (ii) with a maximum spatial separation of 300 km, and (iii) acquired within $\pm 2$ hours. Due
to the dense nature of the OCO-2 soundings relative to the sparseness of the GOSAT soundings, there are typically between
zero and several hundred matched OCO-2 soundings per GOSAT footprint. A lower limit of 10 and an upper limit of 100
(randomly selected) OCO-2 soundings that meet the coincidence criteria were set in order to retain the GOSAT sounding for
analysis. The individual L2FP quality flags are applied for both GOSAT and OCO-2 during the collocation procedure, and then
the mean value of $XCO_2$ from the 10 to 100 collocated OCO-2 soundings is calculated and subtracted from the corresponding
GOSAT $XCO_2$ to produce $\Delta XCO_2^{OCO\text{-}2}$.

Here we compare ACOS GOSAT v9 against OCO-2 v10 (rather than to the deprecated v9), since we assume that science users
will adopt the newest OCO-2 product. Major updates to the version 10 ACOS L2FP algorithm (Osterman et al., 2020) include
(i) an upgrade of ABSCO spectroscopic parameters from v5.0 to v5.1 (Payne et al., 2020), (ii) an improved solar continuum
model, (iii) improved aerosol priors using GEOS5-FP-IT daily means with tighter constraints (Nelson and O'Dell, 2019), (iv)
updated $CO_2$ priors similar to the forthcoming TCCON ggg2020 values, (v) a quadratic fit for land surface albedos, and (vi) a
loosened constraint for the solar induced chlorophyll fluorescence prior. These differences were summarized in Section 3.1.

To account for the use of different $CO_2$ priors in the v9 and v10 L2FP code, a correction was made to the ACOS GOSAT v9
$XCO_2$ as follows;

$$XCO'_2 = XCO_2 + \sum_{i=1}^{i=20} h_i(1-a_i) \cdot (u'_{a,i} - u_{a,i}), \tag{1}$$

where the second term on the right provides the perturbation ($\delta XCO_2$) to the bias corrected $XCO_2$. Here, $h$ is the $XCO_2$ pres-
sure weighting function, $a$ is the normalized $XCO_2$ averaging kernel, $u_a$ is the ACOS v9 $CO_2$ prior profile used for GOSAT,
and $u'_a$ is the ACOS v10 $CO_2$ prior profile used for OCO-2. The summation takes place over the 20 vertical levels defined in
the L2FP code. In summary, the total adjustment to the ACOS GOSAT $XCO_2$ value is calculated as the contribution of the
difference in the vertical $CO_2$ priors at each level weighted by the averaging kernel. The global mean adjustment due to the
$CO_2$ prior correction was approximately 0.2 ppm, with 95% of corrections between -0.1 and +0.5 ppm..

Spatial maps of $\Delta XCO_2^{OCO\text{-}2}$ (GOSAT v9 - OCO-2 v10) for the prior-corrected, collocated soundings are shown in Figures 13
and 14 for land and ocean, respectively. In each figure the maps are shown by season at $2.5°$ latitude by $5°$ longitude reso-
lution. In all seasons, higher scatter in $\Delta XCO_2^{OCO\text{-}2}$ is observed over land ($\simeq 1$ ppm) than over ocean ($< 0.7$ ppm), likely due
to variability of land surface features and/or lower signal-to-noise ratios of the radiance measurements. The annual global
mean $\Delta XCO_2^{OCO\text{-}2}$ for land is near zero (0.06 ppm), and exhibits little variation with season. For Ocean-Glint, the global mean
$\Delta XCO_2^{OCO\text{-}2}$ is larger (-0.40 ppm) and varies more significantly by season from -0.2 (DJF) to -0.6 ppm (JJA). The disagree-
ments in the Ocean-Glint data tend to be spatially coherent, with a notably large negative difference in the NH in most seasons.



Currently, the underlying cause of these disagreements is unknown, and could stem from instrument calibration or sampling related issues, differences in retrieval algorithm versions, or even collocation issues.

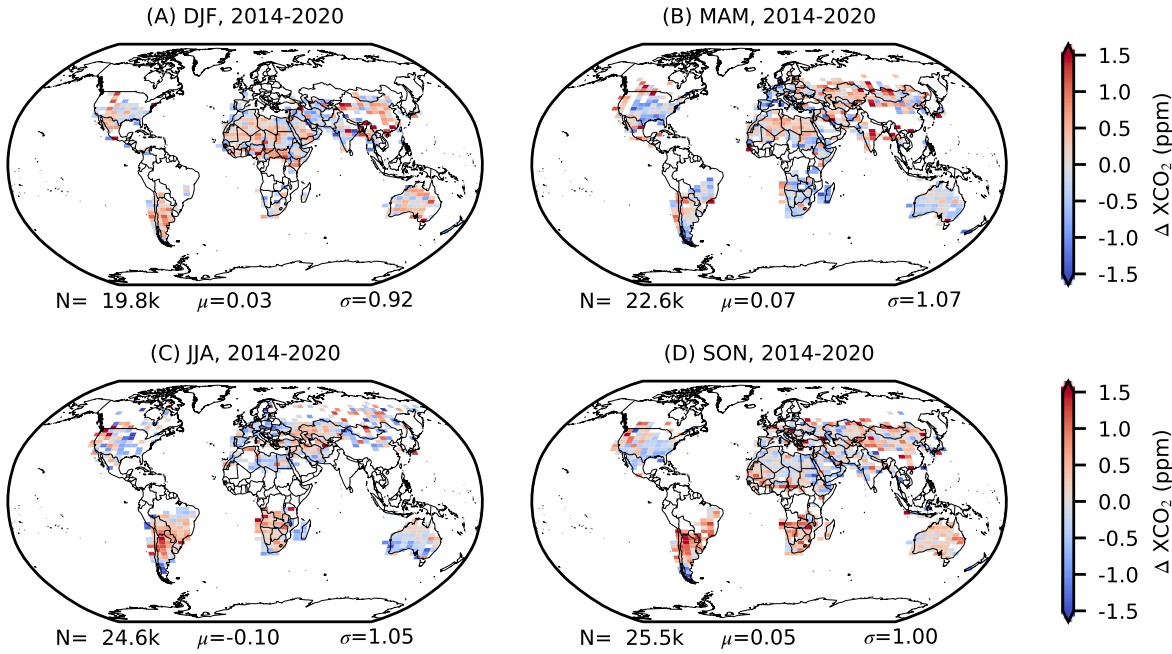

**Figure 13.** Spatial distribution of the bias corrected $\Delta XCO_2^{OCO\text{-}2}$ (GOSAT v9 minus OCO-2 v10) for the good QF land soundings for DJF (A), MAM (B), JJA (C), and SON (D) for the overlapping period August 2014 through June 2020. The spatio-temporal requirements for matched soundings are a maximum separation of 300 km, and observation time within $\pm 2$ hours. Maps are gridded at 2.5° latitude by 5° longitude resolution, and only grid boxes with at least 10 matched soundings are shown.

The disagreement in $XCO_2$ for Ocean-Glint between ACOS GOSAT v9 and OCO-2 v10 is highlighted in Panel A of Figure 15, which shows $\Delta XCO_2^{OCO\text{-}2}$ for the period September 2014 through December 2020 at 30 day by 15° latitude resolution for the Ocean-Glint observations. Panel B shows the number of collocated soundings in each bin. A large SH positive difference in $\Delta XCO_2^{OCO\text{-}2}$ (GOSAT higher than OCO-2 by $\simeq 0.5$ ppm) is observed for the first two years of the time record. Then, in early 2016, there is what appears to be an abrupt jump to a large negative difference (GOSAT lower than OCO-2 by $\simeq$ -0.5 ppm)
in the NH. From this point forward, $\Delta XCO_2^{OCO\text{-}2}$ appears to be reasonably stable in time, although there is a persistent low difference in the NH for the remainder of the record.



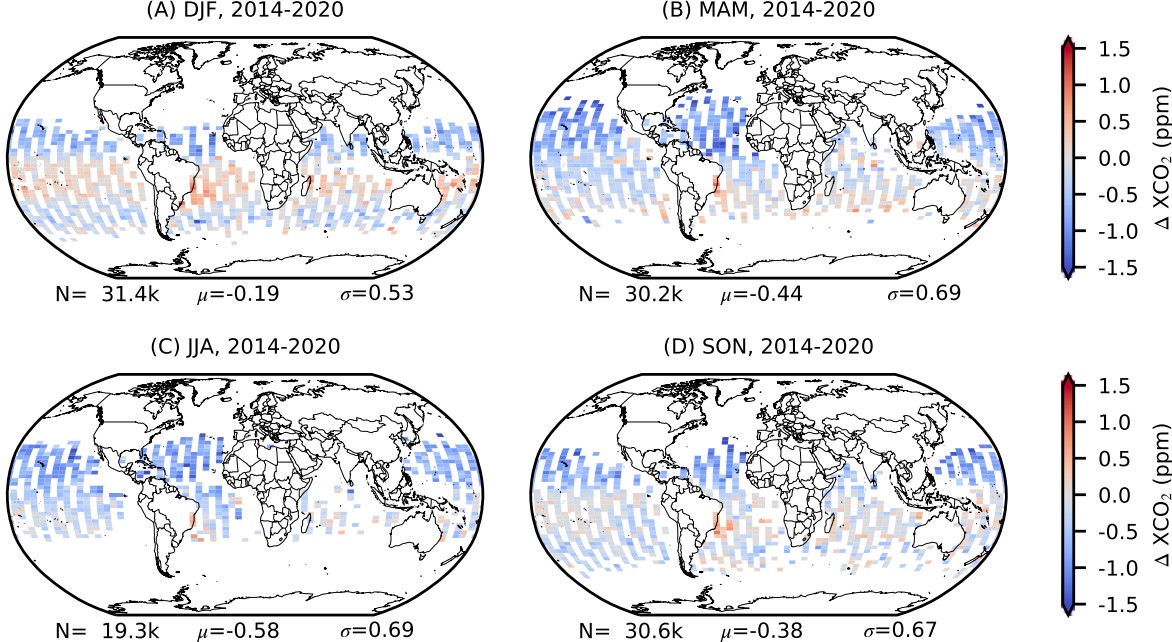

**Figure 14.** Same as Figure 13, but for Ocean-Glint observations.

Figure 16 shows the $\Delta \text{XCO}_2^{\text{OCO-2}}$ data for the combined land H-gain and M-gain data, similar to Figure 15. The main feature here is that the overall variability is larger compared to the Ocean-Glint data, which we attribute to biases introduced by variations in both topography and surface albedo. A slightly positive (red) signal is observed during the September to December months in the SH, especially in 2014, 2018, and 2019. Although, additional investigation into such signals is warranted, it is beyond the scope of the current work.

## 5   Summary

The v9 ACOS GOSAT $\text{XCO}_2$ product, spanning February 2009 through June 2020, has been compared to $\text{XCO}_2$ estimates from TCCON, a suite of atmospheric inversion systems (models), and with collocated OCO-2 v10 data. The ACOS GOSAT v9 product is an improvement over ACOS GOSAT v7.3 relative to these standards. The v9 product provides a significant extension of the data record and contains data in M-gain viewing mode over bright land surfaces.

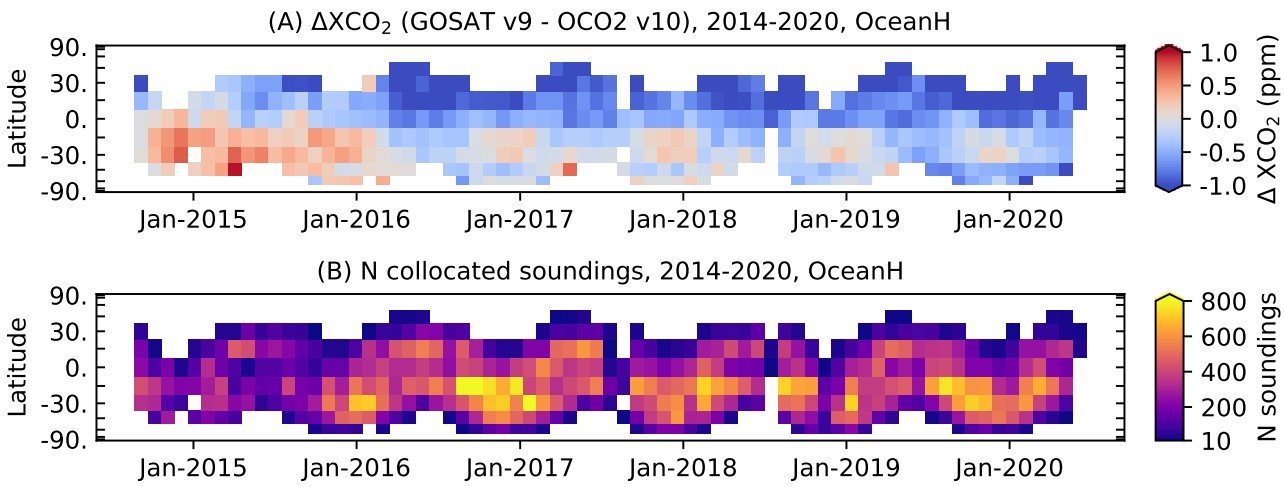

**Figure 15.** Difference in XCO$_2$ between ACOS GOSAT v9 and OCO-2 v10 ($\Delta$XCO$_2^{\text{OCO-2}}$) as a function of time and latitude at 30 day by 15° latitude resolution for Ocean-Glint observations (panel A). Panel (B) shows the sounding density of the collocated soundings. Grid cells containing less than 10 collocations are colored white.

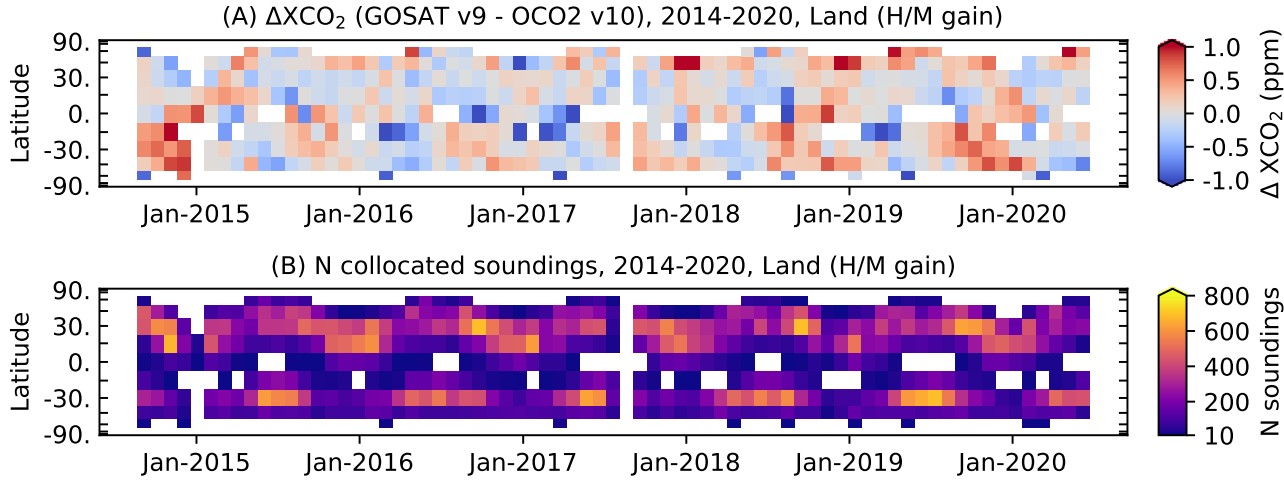

**Figure 16.** Same as Figure 15, but for land observations (combined H-gain and M-gain).



Of the 37.4 M estimates of $XCO_2$ contained in the ACOS GOSAT v9 data record, approximately 80% were prefiltered due to contamination by cloud and/or aerosol, or due to insufficient SNR. Of the 7.0 M that were selected to run through the ACOS L2FP algorithm, approximately 6.1 M returned valid estimates of $XCO_2$. However, only $\simeq 2$ M of those were identified as being of "good" quality. This represents 5.4% of the total recorded soundings. The quality filtering and bias correction variables used for ACOS GOSAT v9 were similar to those used in previous product versions, and similar to those used for OCO-2 v9

and v10, but include for the first time, a correction to account for a small temporal drift in the data.

Comparisons with collocated estimates of $XCO_2$ from TCCON indicate overpass mean biases of $\simeq 0.1$ to 0.2 ppm, and standard deviations of $\simeq 1.5$ ppm. The mean squared error against TCCON is highest for land observations in the northern mid-latitudes $(30 - 60°N)$, and lowest for Ocean-Glint and SH land M-gain observations. The statistics show improvement when compared

to the results for v7.3, which spanned a shorter time period (April 2009 to June 2016). Specifically, the standard deviation of the mean station bias for the 26 sites is 0.41 ppm for the ACOS GOSAT v9 record, compared to 0.51 ppm at 23 stations for ACOS GOSAT v7.3.

Comparisons with collocated $XCO_2$ derived from a suite of 4 atmospheric inversion systems (models) suggests annual global

mean differences of $\simeq 0.15$ ppm and standard deviation of $\simeq 0.5$ ppm. Hemispherical differences in $XCO_2$ estimates over oceans were observed, as well as robust subcontinental scale land features. Results indicate better agreement with models in the ACOS v9 product ($\mu$=-0.20 ppm, $\sigma$=0.8 ppm) compared to v7.3 ($\mu$=-0.54 ppm, $\sigma$=1.0 ppm) for the overlapping period April 2009 through June 2016.

Comparisons with collocated OCO-2 v10 $XCO_2$ data show low bias but relatively high scatter for land observations ($\mu$=0.06 ppm, $\sigma$=1.0 ppm). Increased scatter over land is expected due to $XCO_2$ bias introduced by variability in topography and surface albedo. However, for Ocean-Glint observations, although the $XCO_2$ scatter is lower than that for land as expected ($\sigma$=0.7 ppm), the global mean bias is relatively high ($\mu$=-0.4 ppm), with unaccounted for hemispheric and time differences. These are issues that must be resolved in order for GOSAT v9 and OCO-2 v10 data to provide consistent information to atmospheric inversion

systems for assessing fluxes of $CO_2$.

Global estimates of $CO_2$ derived from satellite measurements provide coverage in traditionally data sparse regions where ground-based measurements are difficult. The assimilation of satellite $XCO_2$ into atmospheric inversion systems to quantify the spatiotemporal variations of carbon fluxes is a promising, but challenging, area of research. This research continues to

benefit from various improvements in transport models, atmospheric inversion systems, and satellite retrievals. The role of the GOSAT record in this field remains unique due to its exceptional 11 year length and its coverage of nearly 5.5 years of the carbon cycle prior to the launch of OCO-2. The ACOS GOSAT v9 L2Std and L2Lite file products are both available on the NASA GES DISC (OCO-2 Science Team et al., 2019b, a).



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

**Data availability**

The ACOS GOSAT v9 $XCO_2$ data are publicly available via NASA's GES-DISC at https://doi.org/10.5067/OSGTIL9OV0PN (L2Std files, OCO-2 Science Team et al. (2019b)) and https://doi.org/10.5067/VWSABTO7ZII4 (L2Lite files, OCO-2 Science Team et al. (2019a)). The OCO-2 v10 L2Lite files containing the bias corrected and quality filtered $XCO_2$ data are avail-
able on the GES-DISC at https://doi.org/10.5067/E4E140XDMPO2 (OCO-2 Science Team et al., 2020). The TCCON data for individual stations are available on the CaltechDATA site at https://data.caltech.edu/. The CarbonTracker data are available on the NOAA GML site at https://carbontracker.noaa.gov. The Carboscope model data are available at http://www.bgc-jena.mpg.de/CarboScope. The CAMS model data are available at https://atmosphere.copernicus.eu/data. The UoL model data are available at https://www.geos.ed.ac.uk/ lfeng/.

*Author contributions.* TET and CWO conceptualized the study and performed formal analysis. Additional formal analysis was performed by HL and AC. Funding acquisition was provided by DC, AK, POW, MG, and AE. Algorithm development was provided by TET, CWO, DC, POW, MG, AE, BF, MK, RRN, AM, and GO. Model data was provided by FC, PIP, and LF. TCCON stations were maintained and data provided by POW, NMD, MKD, DGF, OEG, DWTG, FH, LTI, RK, CL, MDM, IM, JN, YSO, HO, DFP, MR, CMR, MS, MKS, KS, KS, RS, YT, VAV, MW, TW, and DW. Paper writing of the draft was performed by TET, CWO, and DC. All authors contributed to editing the
final version of the manuscript.

*Competing interests.* We declare no competing interests in this work.

*Acknowledgements.* The CSU contribution to this work was supported by JPL subcontract 1439002. TET acknowledges assistance from Peter Somkuti and Heather Cronk at CSU/CIRA with Python map plotting. CarbonTracker results provided by NOAA ESRL, Boulder, Colorado, USA from the website at http://carbontracker.noaa.gov. The GEOS data used in this study were provided by the Global Modeling
and Assimilation Office (GMAO) at NASA Goddard Space Flight Center. HL acknowledges support from the Academy of Finland (project 331829). AM's contributions to this work were supported by JPL subcontract 1577173. P.I.P. and L.F. acknowledge support from the UK National Centre for Earth Observation funded the National Environment Research Council (NE/R016518/1). The TCCON stations at Rikubetsu, Tsukuba, and Burgos are supported in part by the GOSAT series project. Local support for Burgos is provided by the Energy Development Corporation (EDC, Philippines). The TCCON site at Réunion Island is operated by the Royal Belgian Institute for Space Aeronomy with
financial support since 2014 by the EU project ICOS-Inwire and the ministerial decree for ICOS (FR/35/IC1 to FR/35/C6) and local activities supported by LACy/UMR8105 and by OSU-R/UMS3365 – Université de La Réunion. The TCCON stations at Garmisch and Zugspitze



have been supported by the European Space Agency (ESA) under grant 4000120088/17/I-EF and by the German Bundesministerium für Wirtschaft und Energie (BMWi) via the DLR under grant 50EE1711D as well as by the Helmholtz Society via the research programme ATMO. The Paris TCCON site has received funding from Sorbonne Université, the French research center CNRS, the French space agency

CNES, and Région Île-de-France. The Ascension Island TCCON station has been supported by the European Space Agency (ESA) under grant 4000120088/17/I-EF and by the German Bundesministerium für Wirtschaft und Energie (BMWi) under grants 50EE1711C and 50EE1711E. We thank the ESA Ariane Tracking Station at North East Bay, Ascension Island, for hosting and local support. The Anmywondo TCCON station was funded by the Korea Meteorological Administration Research and Development Program "Development of Monitoring and Analysis Techniques for Atmospheric Composition in Korea." under Grant (KMA 2018-00522) NMD is supported by an Australian

Research Council (ARC) Future Fellowship, FT180100327. The Darwin and Wollongong TCCON sites have been supported by a series of ARC grants, including DP160100598, DP140100552, DP110103118, DP0879468 and LE0668470, and NASA grants NAG5-12247 and NNG05-GD07G. The Eureka measurements were made at the Polar Environment Atmospheric Research Laboratory (PEARL) by the Canadian Network for the Detection of Atmospheric Change (CANDAC), primarily supported by the Natural Sciences and Engineering Research Council of Canada, Environment and Climate Change Canada, and the Canadian Space Agency. MKD thanks the LANL LDRD program

for support operating the Four Corners TCCON site. The TCCON Nicosia site has received additional support from the European Union's Horizon 2020 research and innovation programme under grant agreement No. 856612 and the Cyprus Government, and by the University of Bremen. The Anmyeondo TCCON station is funded by the Korea Meteorological Administration Research and Development Program "Development of Monitoring and Analysis Techniques for Atmospheric Composition in Korea." under Grant (KMA 2018-00522).



## Appendix A: Seasonal cycle plots of ACOS GOSAT v9 $XCO_2$ versus TCCON for individual stations

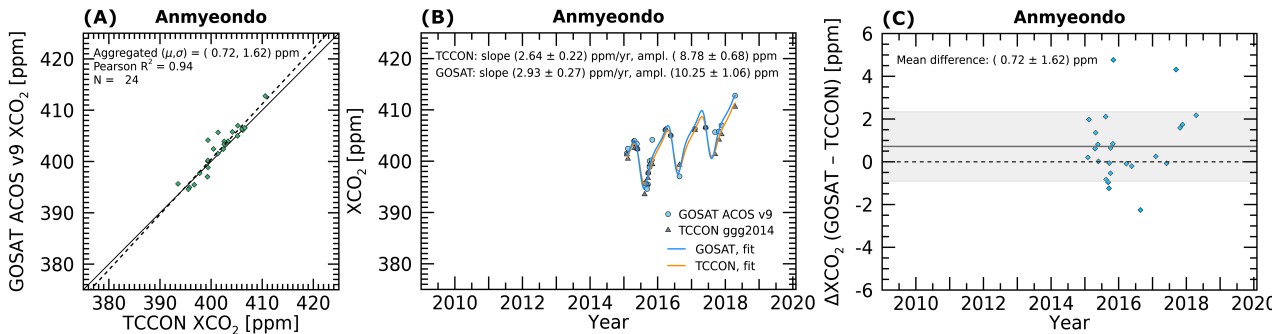

**Figure A1.** Daily averaged bias-corrected ACOS GOSAT v9 versus collocated TCCON $XCO_2$ at Anmyeondo, Korea. Left panel (A) shows the one-to-one scatter plot of the daily mean $XCO_2$ for GOSAT versus TCCON. Middle panel (B) shows the time series of daily mean GOSAT $XCO_2$ (blue circles) with fit (blue line) and the TCCON $XCO_2$ (grey triangles) with fit (orange line) over the eleven year data record. Right panel (C) shows the time series of calculated $\Delta XCO_2^{TCCON}$, with the mean difference (horizontal solid black line) and $\pm 1$ standard deviation (gray shading). In these plots, the three GOSAT observation modes have been combined in order to provide the maximum number of collocations possible for the seasonal fits.

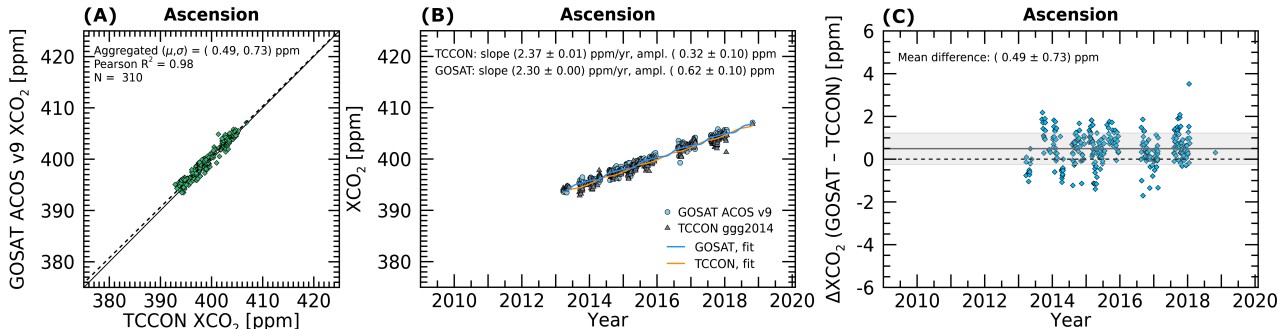

**Figure A2.** Same as Figure A1, but for Ascension Island, located in the Pacific ocean off the west coast of Africa.





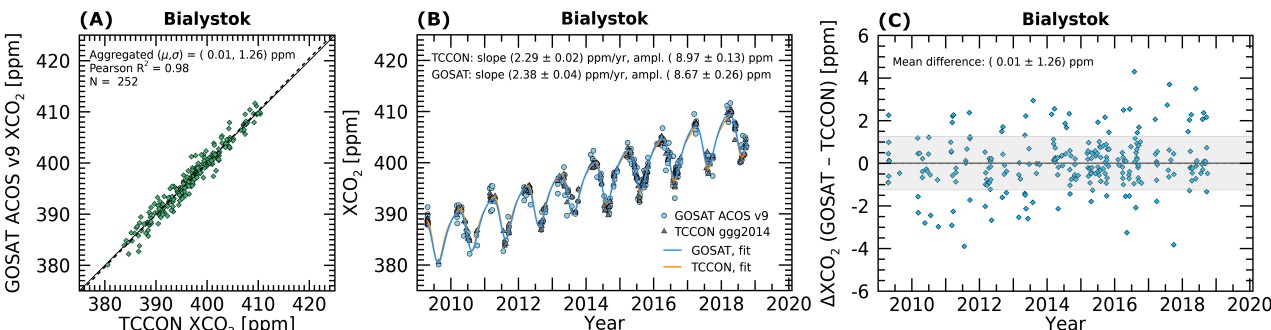

**Figure A3.** Same as Figure A1, but for Białystok, Poland.

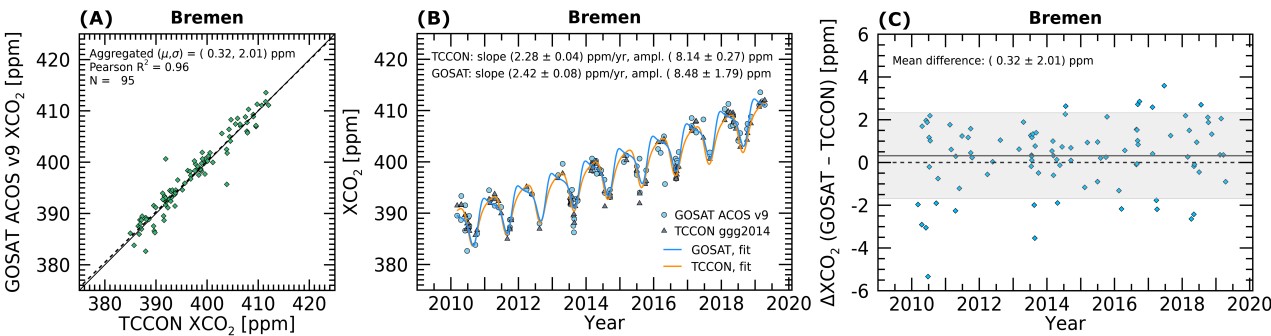

**Figure A4.** Same as Figure A1, but for Bremen, Germany.

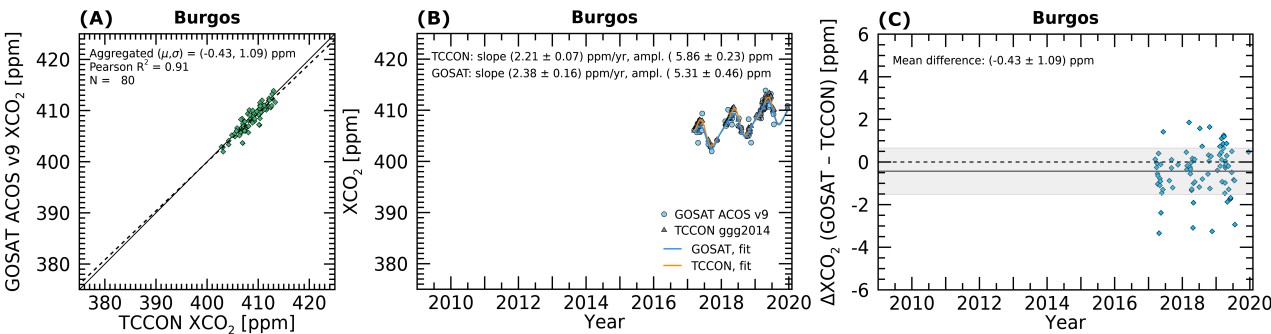

**Figure A5.** Same as Figure A1, but for Burgos, Philippines.




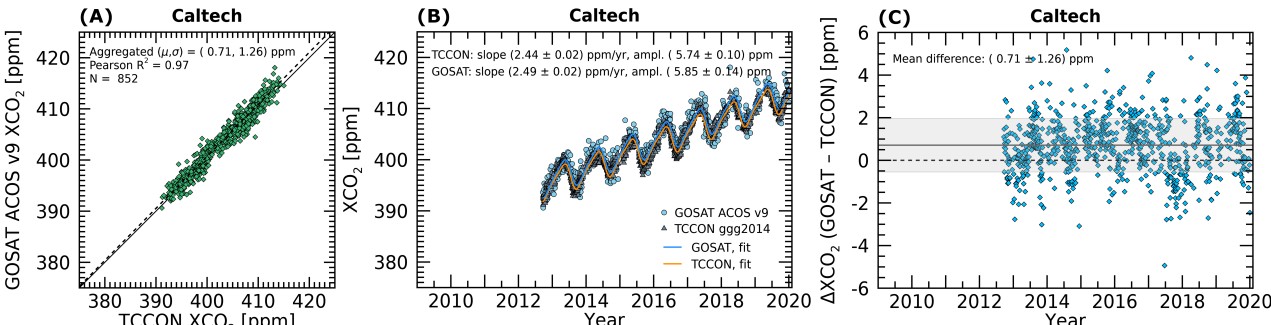

**Figure A6.** Same as Figure A1, but for Caltech, California.

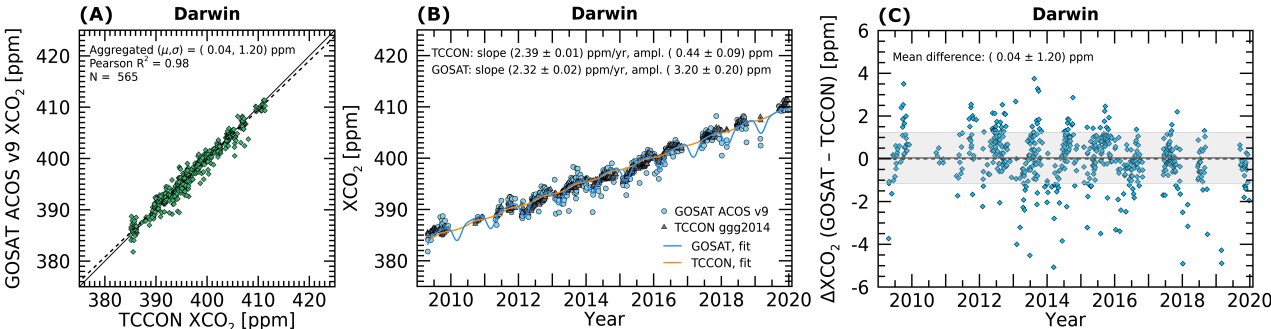

**Figure A7.** Same as Figure A1, but for Darwin, Australia.

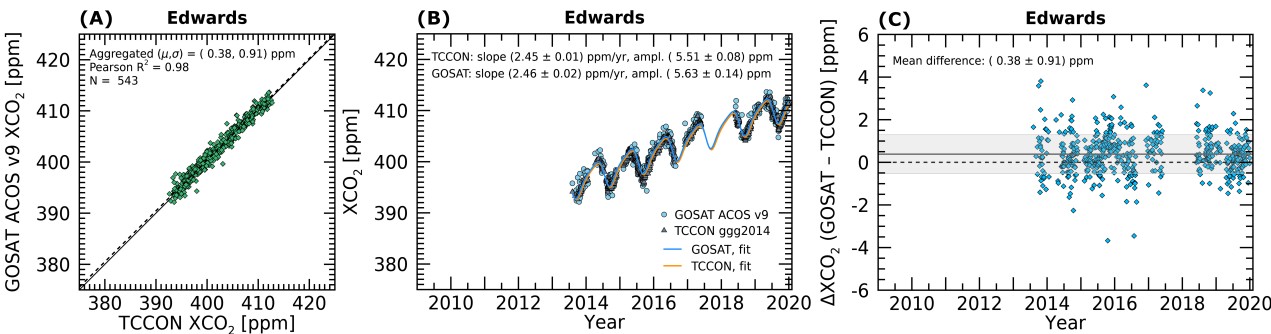

**Figure A8.** Same as Figure A1, but for Edwards, California.



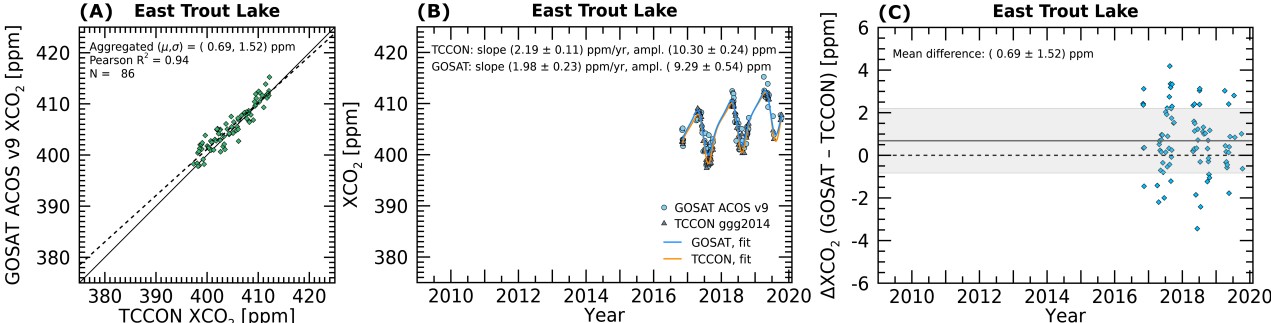

**Figure A9.** Same as Figure A1, but for East Trout Lake, Canada.

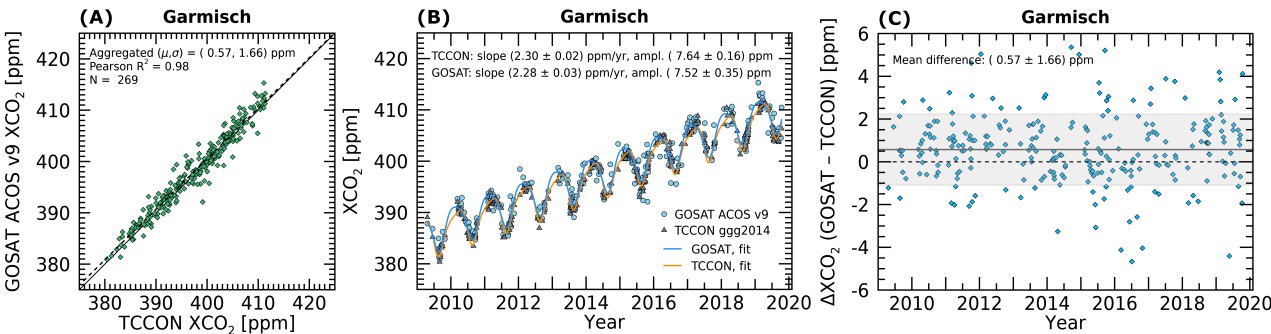

**Figure A10.** Same as Figure A1, but for Garmisch, Germany.

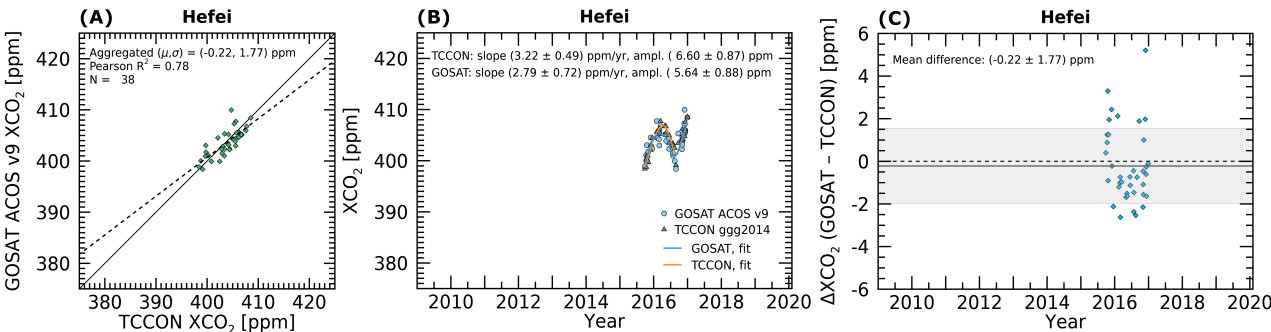

**Figure A11.** Same as Figure A1, but for Hefei, China.




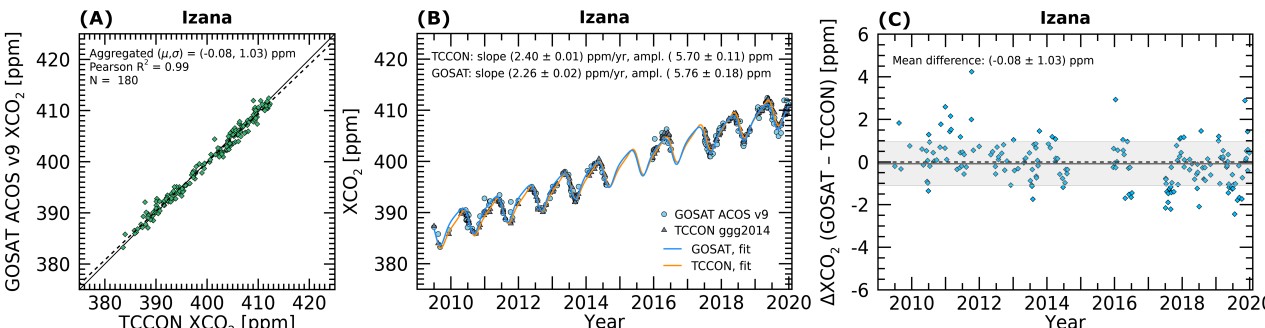

**Figure A12.** Same as Figure A1, but for Izaña, Tenerife, Spain.

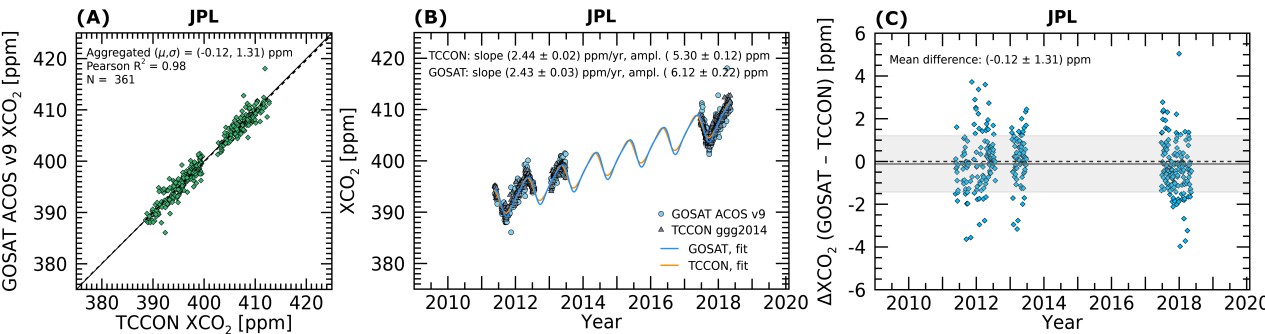

**Figure A13.** Same as Figure A1, but for Jet Propulsion Laboratory (JPL), Pasadena, California. This site has been used occasionally for the simultaneous operation of a TCCON instrument during the thermal vacuum testing of OCO-2 (Frankenberg et al., 2015) and OCO-3.

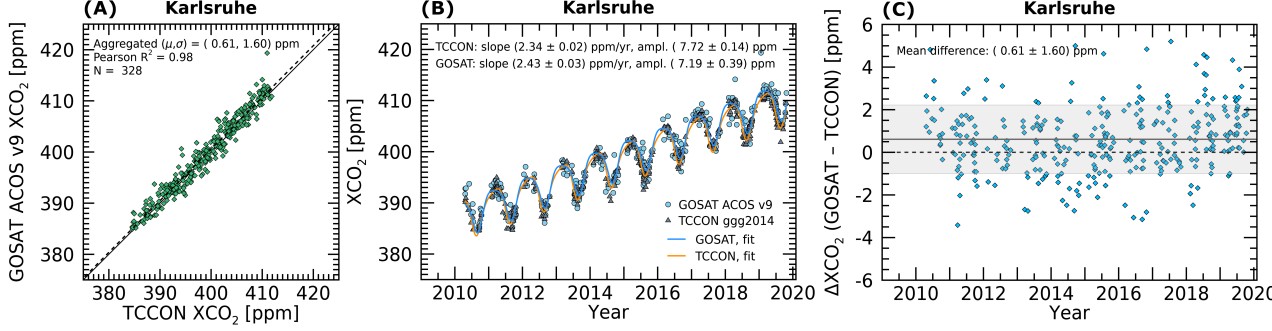

**Figure A14.** Same as Figure A1, but for Karlsruhe, Germany.





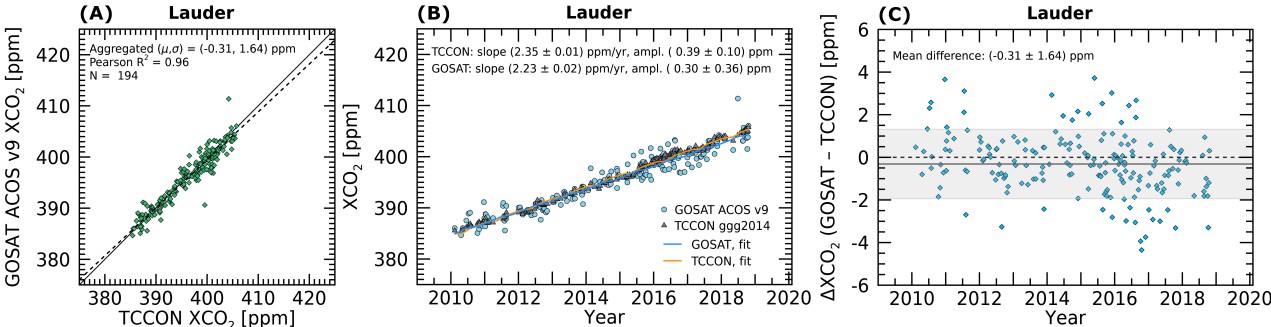

**Figure A15.** Same as Figure A1, but for Lauder, New Zealand.

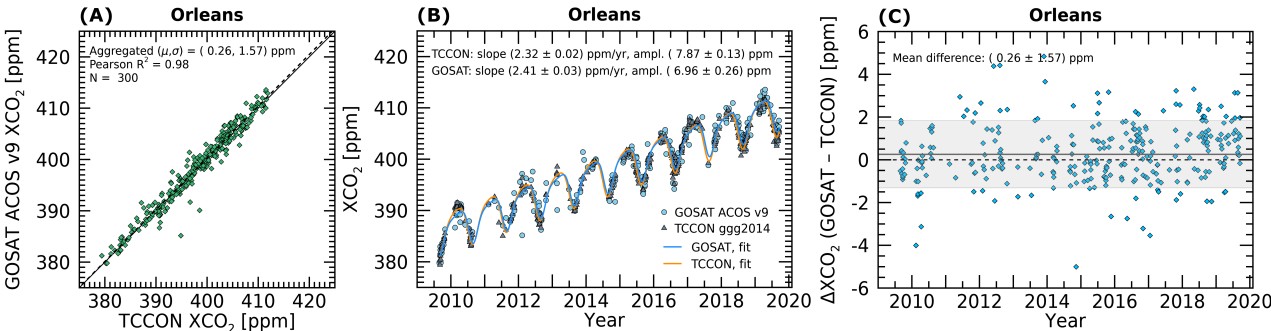

**Figure A16.** Same as Figure A1, but for Orleans, France.

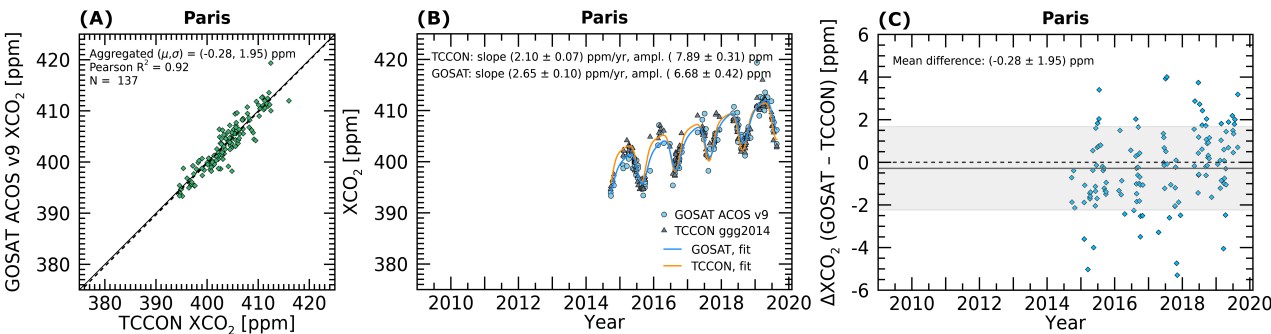

**Figure A17.** Same as Figure A1, but for Paris, France.





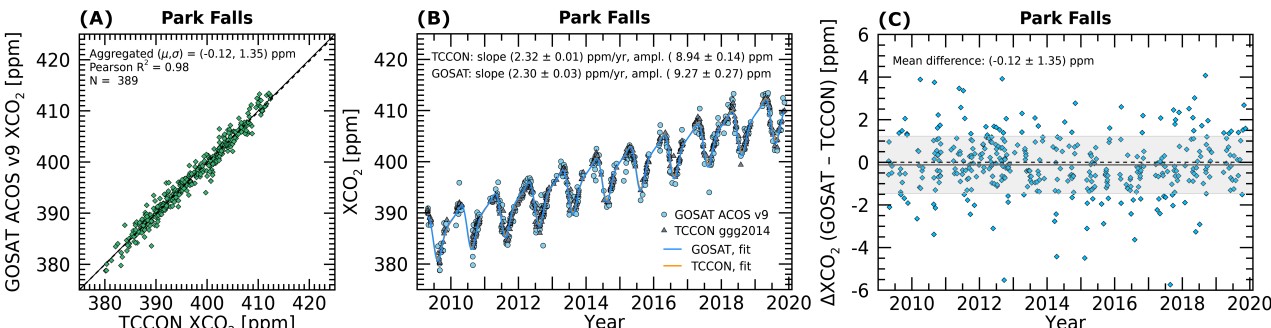

**Figure A18.** Same as Figure A1, but for Park Falls, Wisconsin.

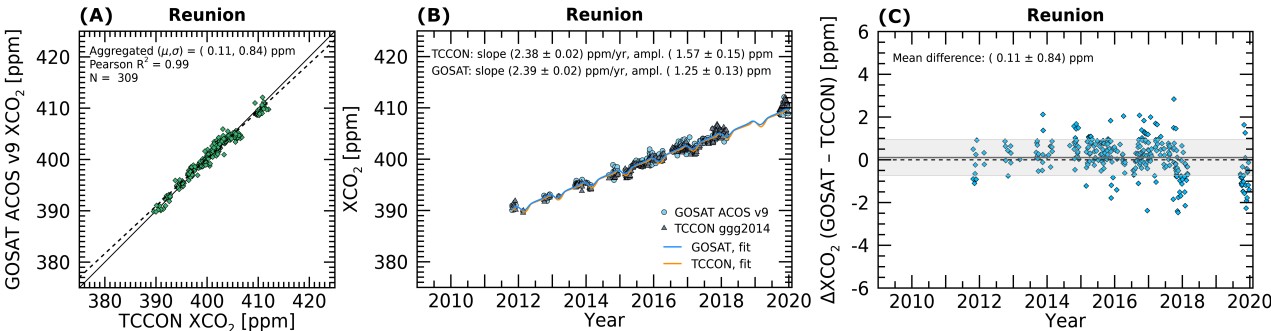

**Figure A19.** Same as Figure A1, but for Reunion Island, off the east coast of Madagascar.

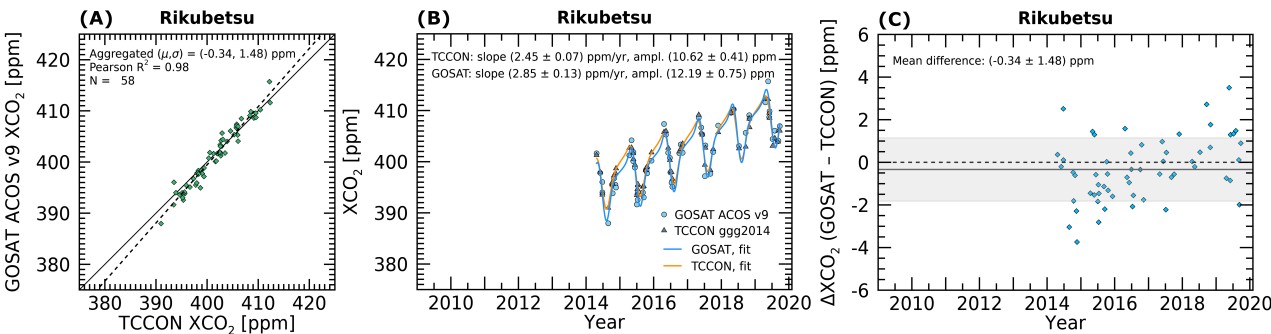

**Figure A20.** Same as Figure A1, but for Rikibitsu, Japan.





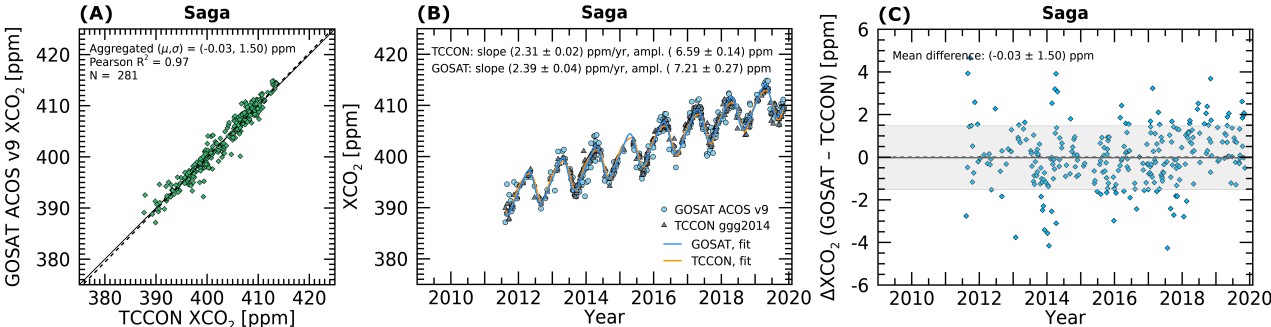

**Figure A21.** Same as Figure A1, but for Saga, Japan.

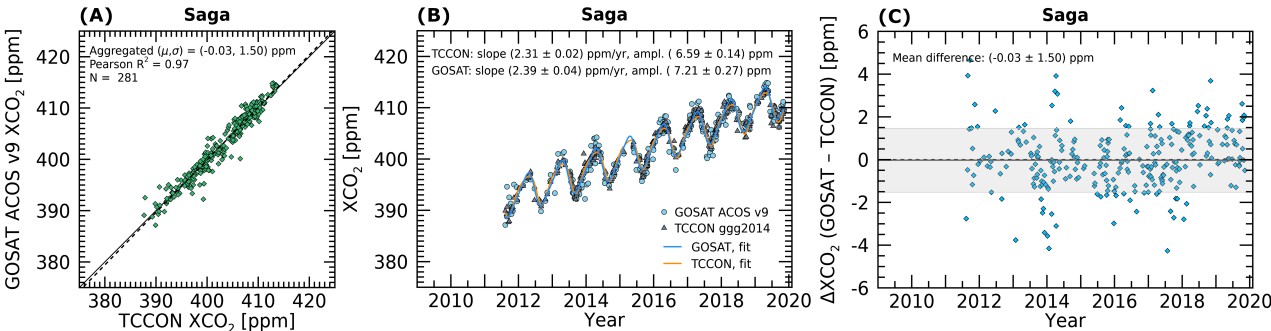

**Figure A22.** Same as Figure A1, but for Sodanyla, Finland.

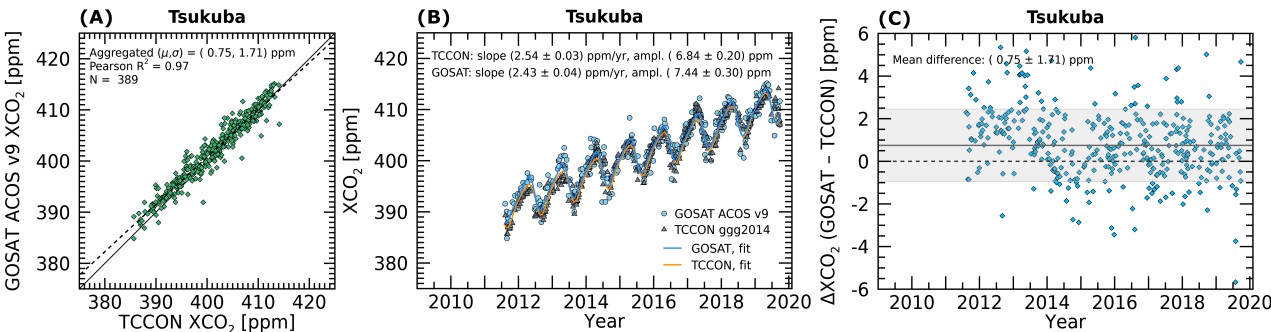

**Figure A23.** Same as Figure A1, but for Tskuba, Japan.





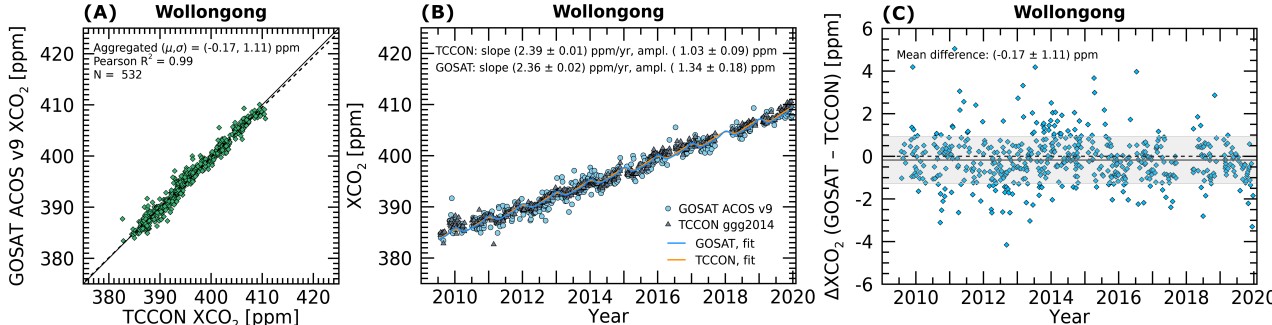

**Figure A24.** Same as Figure A1, but for Wollongong, Australia.

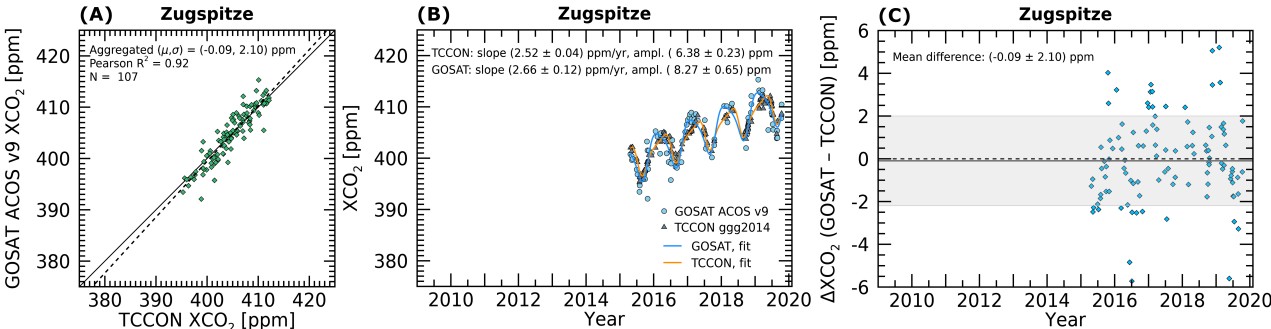

**Figure A25.** Same as Figure A1, but for Zugspitze, Germany.