# Peer review of "An eleven year record of XCO2 estimates derived from GOSAT measurements using the NASA ACOS version 9 retrieval algorithm"

_Earth System Science Data, 2021_

## Author Comment (AC1)

**Response to Reviewer #1**
**An eleven year record of $XCO_2$ estimates derived from GOSAT measurements using the NASA ACOS version 9 retrieval algorithm**

T.E. Taylor et al.

Thank you to the reviewer for the suggestions and comments. We appreciate your time and concern. We have addressed each enumerated point below. The original reviewer comment is given in black. Our reply is given in blue. Modifications to the manuscript text are given in red.

5    1. Figure 5: It would be interesting to add a column c) ACOSv9- ACOSv7 to also see the spatial and temporal differences of the retrieval versions.

This was an interesting and engaging suggestion. To do this we first "sounding matched" the v7.3 and v9 data sets and screened both for good QF. This truncates the record to span April 2009 to June 2016, and removes Land M-gain observations as these were not present in v7.3. The matching procedure resulted in a total of 311 k and 398 k soundings
10    for Land and Ocean-Glint observations, respectively. We then calculated $\Delta XCO_2$ on a sounding-by-sounding basis and produced histograms, spatially gridded maps, and Hovmoller-like plots ($\Delta XCO_2$ versus time and latitude) by observation mode. This information is presented in what is now Figure 7. The most prominent feature is a drift of the $\Delta XCO_2$ signal in time. This feature can be partially accounted for with the time dependent bias correction parameter implemented in v9 of 0.05 ppm/yr (Land) and 0.1 ppm/yr (Ocean-Glint), although the remainder of the feature has unknown source.

15    This direct analysis between v9 and v7.3 is interesting and fits well with the scope of the research. We have therefore elected to add this substantial new material to the paper, including the new figure. We note however, that this direct comparison does not allow for determination as to which product is closer to truth. But, the analysis of both v9 and v7.3 against truth metrics (TCCON and models) indicates that the v9 product is an improvement over v7.3, as already described in the original manuscript.

20    The following discussion and figure have been added to Section 4.2.

A quantification of differences in the bias corrected ACOS GOSAT v9 $XCO_2$ data product relative to the v7.3 product is given in Figure 1 [actually will be Figure 7 in the manuscript] for the overlapping period. The top row (panels A through C) show results for the Land H-gain observations, while the lower row (panels D through F) show results for the Ocean-Glint observations. Only soundings that were present in both data sets and assigned a good quality L2FP flag were used
25    in this comparison. This restricts the analysis to April 2009 through June 2016, and also eliminates the v9 Land M-gain data, as no M-gain data exists in the v7.3 product. The mean and standard deviation of the $\Delta XCO_2$ for the approximately

311 k Land soundings at the single sounding level are -0.18 ppm and 0.72 ppm, respectively, as shown in Panel (A). When gridded and mapped at 2.5° latitude by 5° longitude resolution, as shown in Panel (B), the majority of the negative signal (v9 $XCO_2$ lower than v7.3) occurs at latitudes greater than approximately 45°. Most of the land mass at latitudes less than 45° have $\Delta XCO_2$ values closer to zero, with the largest positive signals appearing over equatorial forests. Furthermore, when the data is gridded and viewed versus time and latitude in 30 d by 15° increments, respectively, as in Panel (C), we see that the $\Delta XCO_2$ signal has an increasing tendency in time, i.e., the v9 $XCO_2$ increases more rapidly in time than v7.3. The cause of this effect is currently unknown, but is partially due to the implementation in v9 of a time dependent bias correction term of +0.05 ppm/yr for Land observations. This translates into an expected change of about 0.35 ppm in the v9 record over the 2009 to 2016 time span.

For the Ocean-Glint observations at the single sounding level (Panel D), the mean and standard deviation of the $\Delta XCO_2$ are +0.28 ppm and 0.79 ppm, respectively. When gridded and mapped at 2.5° latitude by 5° longitude resolution (Panel E), the spatial distribution is fairly smooth, i.e., low variation in both latitude and longitude. Finally, when the data is gridded versus time and latitude (Panel F), the modest variation in latitude is confirmed, but a substantial time dependence is observed, with the $\Delta XCO_2$ signal beginning negative in 2009 (v9 $XCO_2$ lower than v7.3), and switching to a positive $\Delta XCO_2$ signal by 2016 (v9 $XCO_2$ higher than v7.3). The time dependent bias correction term for Ocean-Glint observations was +0.1 ppm/yr. This translates into an expected change of about 0.7 ppm over the 2009 to 2016 time span in the v9 record, accounting for some, but not all of this time dependent difference between v9 and v7.3.

This direct comparison between the v9 and v7.3 $XCO_2$ product only allows for statements as to their differences. It does not allow one to deduce which is closer to truth. Therefore, an analysis of the v9 $XCO_2$ data product against truth metrics follows. Furthermore, it is difficult to accurately determine the effect that the new v9 $XCO_2$ product will have on atmospheric inversion system results relative to v7.3 without further detailed study.

2. Fig 10: You explain the filter criteria of the MMM (L239f). As the models will deviate from each other more where prior uncertainties are high and assimilating data coverage is low, the models will deviate more in remote regions. Thus, data in remote regions will be rejected for quality assessment. Therefore, a map showing the data density of co-located samples would be useful to the reader. If filtered data coverage of MMM in remote regions is small, also a discussion of how this might influence/limit the quality assessment should be given.

As the native model fields are continuous in time and space, the data density of the collocated samples is driven by the spatial distribution of the Good QF GOSAT soundings, which is shown in Figure 1D in the manuscript. The question becomes, what fraction (and spatial distribution) of the good QF GOSAT soundings do not have valid MMM, and for what reason. Here, we present maps showing the spatial distribution of the rejected soundings, shown in Figures 2 and 3. We see that approximately 17% of the GOSAT soundings with a Good QF do not have a valid MMM for comparison due to rejection by one or more of the model criteria, and that the rejections are in fact spatially coherent. Although this additional material on the models could be tidied up a bit and made into an appendix (at the reviewers discretion), we suggest that it should not be added directly into the manuscript to avoid lengthening the main document.

[Figure]

**Figure 1.** Analysis of the ACOS GOSAT calculated $\Delta XCO_2$ (v9 minus v7.3) for the quality filtered and bias corrected soundings for the overlapping period spanning April 2009 through May 2016. Panels (A) through (C) show results for the Land H-gain observations, while Panels (D) through (F) show results for Ocean-Glint observations. Panels (A) and (D) shows the single sounding frequency distribution of $\Delta XCO_2$ in 0.1 ppm bins. Panels (B) and (E) show the spatially gridded $\Delta XCO_2$ at 2.5° latitude by 5° longitude resolution. Panels (C) and (F) show the $\Delta XCO_2$ as a function of time and latitude in 30 d by 15° increments, respectively. The statistics in Panels (A) and (D) were calculated at the single sounding level, while those reported in Panels (B) and (E) were calculated on the grid box means.

Based on the above findings we modified and added to the discussion as follows: Approximately 85% of the GOSAT v9 soundings with a good L2FP quality flag had a valid MMM XCO$_2$ value for analysis. The regions with the highest fraction of rejections occur along the southern ocean (latitude -60°), the Amazon and Congo rain forests, and a broad region across northern Asia.

[Figure]

**Figure 2.** Gridded maps of the sounding density of good quality flagged data and valid model data (top), sounding density of the good QF data that violated one of more model criteria (middle), and fraction of model violations relative to the good QF data.

During this exercise, a minor bug in the GOSAT vs MMM code was found and corrected. Specifically, the threshold for rejecting collocations with less than 3 valid models was incorrectly coded, meaning that approximately 175 k collocations were erroneously being included in the analysis, most of which occurred in year 2018 (because only 2 of the 4 models were present in our archived data set past the end of 2017). This affected Figures 10 and 11 (as numbered in the reviewed manuscript), which have since been updated and are presented here as Figure 4 and Figure 5. The data in the spatial plots are nearly indistinguishable from the originals, but the reported statistics are slightly different. The v9 Ocean-Glint data in the Hovmoller plot now terminates at the end of 2017, instead of 2018. Generally, the agreement of v9 XCO$_2$ with models is now slightly better due to this correction.

65

70

[Figure]

**Figure 3.** Gridded maps of the sounding density (left column) and fraction (right column) for each model selection criteria, relative to the good QF data.

3. L482f: In the manuscript you explain why you use OCO2-v10 and GOSATv9. You make the decision to compare results across satellites AND retrieval versions. However, you then decide to account for parts of the differences in retrieval algorithm (only for different CO2 priors) to compare the products. This is not consistent. I think you should not correct for the different priors used in the comparison as it is not a valid comparison to OCO-2 in any version otherwise. If possible, a short discussion of the effect of the different updates of the retrieval versions would be interesting instead.

We agree that on the surface our adjustment to account for the prior but not account for any other algorithm differences seems inconsistent. However, as discussed in Section 2.5.7 of the DUG, it is recommended that atmospheric inversion modelers always apply an averaging kernel correction. To make this point explicit, the following discussion has been added to Section 3.3 of the data paper:

For each GOSAT sounding, a multi-model median (MMM) $XCO_2$ was calculated from the models having a valid $XCO_2$ estimate for that location and time. Unless otherwise noted, the model $XCO_2$ is taken to be that which a perfect OCO-2

[Figure]

**Figure 4.** Seasonal maps of the mean $\Delta XCO_2^{MMM}$ (GOSAT - MMM) spanning 2009 through 2018 for DJF (A), MAM (B), JJA (C), and SON (D) at 2.5° latitude by 5° longitude resolution. Grid boxes containing less than 10 collocations are colored white.

[Figure]

**Figure 5.** Time series of $\Delta XCO_2^{MMM}$ (ACOS GOSAT v9 - MMM) versus latitude at 30 day by 15° resolution for Ocean-Glint observations for v7.3 (A) and v9 (B). Grid cells containing less than 10 collocations are colored white.

would have observed, $XCO_{2,ak}$; that is, an averaging kernel correction is applied to account for differences between the model profile of $CO_2$ and the ACOS prior in the unmeasured part of the profile:

$$XCO_{2,ak} = \sum_{i=1}^{20} h_i\{a_i u_{m,i} + (1-a_i)u_{a,i}\}, \tag{1}$$

where $h_i$ is the pressure weighting function on the $i = 1...20$ ACOS model levels, $a$ is the normalized ACOS averaging kernel for $CO_2$, $u_m$ is the model profile of $CO_2$, and $u_a$ is the ACOS prior profile of $CO_2$.

Then, in Section 4.5, the discussion has been adjusted as follows:

One complexity in comparing ACOS GOSAT v9 and OCO-2 v10 is the fact that the two versions of the algorithm used different $CO_2$ priors. Typically, models which assimilate satellite $CO_2$ data take into account the unmeasured part of the prior $CO_2$ profile (specified via the retrieval's averaging kernel) via an averaging kernel correction, as given in Eq. 1. Therefore, in order to fairly compare these two data sets as models would assimilate them, we need to remove their difference due to the unmeasured part of the $CO_2$ profile, as follows:

$$XCO_2' = XCO_2 + \sum_{i=1}^{20} h_i(1-a_i) \cdot (u_{a,i}' - u_{a,i}), \tag{2}$$

where $h$ is the $XCO_2$ pressure weighting function, $a$ is the normalized $XCO_2$ averaging kernel, $u_a$ is the ACOS v9 $CO_2$ prior profile used for GOSAT, and $u_a'$ is the ACOS v10 $CO_2$ prior profile used for OCO-2. The summation takes place over the 20 vertical levels defined in the ACOS code. In summary, the total adjustment to the ACOS GOSAT $XCO_2$ value is calculated as the contribution of the difference in the vertical $CO_2$ priors at each level weighted by the one minus the averaging kernel at that level. The global mean adjustment due to the $CO_2$ prior correction was approximately 0.2 ppm, with 95% of corrections between -0.1 and +0.5 ppm.

4. L7: Explicitly state here that no satellite data has been used in the assimilation system here.

   Done.

5. L17: wording "Similarly," does not fit. Do you mean "Further, "?

   Modified the sentence to read; Global mean biases against TCCON and models are less than approximately 0.2 ppm.

6. L71: Why is XCH4 out of the scope of the manuscript? Can you refer the reader to further literature?

   The ACOS algorithm, which was originally developed for retrievals on OCO-2 measurements, has never had the capability to retrieve methane (because OCO does not measure in the methane absorption bands). Based on your comment, and the comment of Reviewer #2, we have decided to remove all instances of methane from the paper in the name of brevity and clarity.

7. L133: L2FP is used, but abbreviation is only introduced in line 139

   The L2FP is defined in the abstact and again at the first use in the Introduction, per the journal guidelines. I also decided to define it again a third time in the opening of Section 3: The ACOS v9 L2FP XCO$_2$ retrieval algorithm for completeness.

8. Table 2: For clarity, you could add superscripts of the respective figures in Fig. 1?

   An additional column was added to the table to identify the corresponding panel in the figure.

9. L204: see above: in-situ assimilating models?

   The sentence now reads: The development of ACOS GOSAT v9 used XCO$_2$ truth metrics derived from both TCCON measurements, and the median CO$_2$ distributions determined from a suite of four atmospheric inversion systems, which do not assimilate satellite CO$_2$ measurements.

10. L214f: What is the reason for the spatial collocation criteria of +/- 2.5 ° lat and +/-5! Lon? Have you performed, or can you reference a footprint analysis here?

    To collocate GOSAT with TCCON, we followed the spatial criteria laid out in Section 4 of [Wunch, AMT, 2017, https://doi.org/10.5194/amt-10-2209-2017]. The sentence now reads: Following the criteria defined in (Wunch, 2017), the spatial collocation criteria for GOSAT soundings were those falling within $\pm 2.5°$ latitude and $\pm 5°$ longitude of a TCCON station for most sites. It is worth keeping in mind that GOSAT samples on any given orbit are order several hundred kilometers apart, i.e., it is quite spatially sparse. Therefore the spatial collocation criteria cannot be too strict. Also, by selecting similar criteria as in earlier work, the validation results are more directly comparable to earlier results.

11. L259: You refer the reader to O'Dell et al. 2020 for details about the bias correction. At least some details on the correction (maybe equation 6 in O'Dell et al., 2018?) would help the understanding and flow of the paper.

    Agreed. The discussion has been modified to read:

    Spurious correlations in the estimates of XCO$_2$ with other retrieval variables due to inadequacies in the modeled physics motivate the application of a bias correction (Wunch, 2011; O'Dell, 2018). Generally such spurious correlations are found with state vector elements such as retrieved surface pressure, various aerosol parameters, and $\delta \nabla_{CO_2}$. For each sensor there are also typically offsets by viewing mode, e.g., land versus ocean-glint, which are accounted for via the bias correction. A general discussion of the ACOS XCO$_2$ bias correction methodology is provided in Section 4 of (O'Dell, 2018), where the fundamental equation is defined as:

$$XCO_{2,bc} = \frac{XCO_{2,raw} - C_P(mode) - C_F(j)}{C_0(mode)}, \tag{3}$$

    where $C_P$ is the mode-dependent parametric bias, $C_F$ is a footprint-dependent bias for footprints 1...8, and $C_0$ represents a mode-dependent global scaling factor. Note that for GOSAT there is no footprint-dependent bias correction term, as is

necessary for OCO due to low level calibration errors across the detector frame. Further, to be consistent with previous ACOS GOSAT data versions, the global divisor is replaced by an additive offset, which is effectively the same because the range of $XCO_2$ variability ($\sim 20\,\text{ppm}$) is small relative to the mean $XCO_2$ ($\sim 400\,\text{ppm}$).

The explicit formula for application of the ACOS GOSAT v9 correction is provided in Section 2.5.6 of the DUG. For both land H-gain and M-gain, a set of five BC variables are used, while Ocean-Glint uses only 3 variables. The difference between the H- and M-gain bias correction over land is minor. New for ACOS GOSAT v9 is the use of a correction against time, which is made possible with an eleven year data record; the corrections are 0.05 ppm/yr over land and 0.10 ppm/yr over water. The source of this spurious drift in the bias-corrected $XCO_2$ is currently unclear and is the subject of further study. Although there is some commonality in the quality filtering and bias correction variables used for ACOS GOSAT v9 (compare Tables 5 and 6), they do differ somewhat, as is typically the case with each sensor and data version.

12. L282: The mean bias should be nearly zero after the bias correction. Why does a median bias persist? Can you add a sentence here?

   I think what you're suggesting is that the mean bias against a truth metric (TCCON or models) should be zero after application of the bias correction. That should be approximately true, and is demonstrated in the $XCO_2$ analysis section. However, what we are showing here in Figure 2 is the actual relative bias correction magnitude as grid box average values. There is no reason why the actual global gridded bias correction should have a mean (or median) of zero, as it represents the adjustment of the $XCO_2$ values to the truth metrics, which could in theory be a large positive or negative number (if the L2FP was performing very poorly). The figure demonstrates that the bias correction is very mode dependent with some minor latitudinal dependence. Further, given that the strongest signal is by mode, and there are unequal numbers of Land H-gain, Land M-gain, and Ocean-Glint observations, we again have no reason to suspect a mean/median near zero.

13. L293: delete "of"

   Done.

14. Figure 7: In the figure caption add where to find the statistics of individual stations.

   I see your point here. While it might make sense to refer the reader to the summary statistics per site given in Table 9, the hangup is that Table 9 is derived from the seasonal cycle analysis, which was performed distinct from the one-to-one all-site-combined analysis. So for example there are a couple of sites presented in Figure 7 that do not appear in Table 9 because there was insufficient data to characterize the seasonal cycle. So we feel like it is best to not make any changes related to this.

15. L382: Any ideas why MAE is a function of latitude?

   Augmented the discussion slightly as follows:

In the v9 Land H-gain data, the MAE is roughly a function of latitude, with the highest values ($\simeq 1.0$ ppm) seen between $60°N - 90°N$, and the lowest values ($\simeq 0.7$ ppm) seen from $30°S - 60°S$. This stands to reason as lower variability of $XCO_2$ in the SH tends to yield better agreement between satellite and ground based observations.

16. L395: Why do you use a more restrictive collocation criteria then the one presented before? This seems inconsistent to me.

   For the main analysis we wanted to retain as many soundings as possible. However, in the extended analysis, the additional criteria are necessary to enable sufficient data for the seasonal cycle fits, e.g. a contiguous year of sampling.

17. L421: delete "time"

   Done.

18. L444: Add a remark what Müller et al. (2021) suggest as it is of high importance for this work as well.

   I have to admit that the citation of Müller et al. (2021) was added quite late in the writing stage. It is an excellent resource and we hope to have the luxury of obtaining that evaluation data for future ACOS development. The discussion was slightly reworded as follows:

   It is unclear why the satellite and models disagree over such large spatial and temporal scales, but recent work by (Müller, 2021) suggests that the ACOS v7.3 (and to a lesser extent v9) $XCO_2$ are in fact biased low by approximately 1 to 1.5 ppm, as compared to a new independent evaluation data set generated from combined ship and aircraft measurements over the open oceans. Further investigation into the source of the ACOS GOSAT biases against models is warranted.

19. L474-479 are repetitive to Section 3.1 and could be deleted.

   Indeed. We have deleted those repetitive sentences.

20. L539: Add sth like " but further investigation is required to explain the remaining disagreement over large spatial and temporal scales."

   Done.

21. L543: What do you mean with "uncounted for hemispheric and time differences"?

   Removed that phrase for clarity, and modified the parenthetical summary statistics for both land and ocean-glint to read: ... for land observations ($\mu$=0.06 ppm, $\sigma$=1.0 ppm, when averaged across seasons)....However, for Ocean-Glint observations, although the $XCO_2$ scatter is lower than that for land as expected ($\sigma$=0.7 ppm), the global mean bias is relatively high ($\mu$=-0.4 ppm, when averaged across seasons).

---

## Author Comment (AC2)

**Response to Reviewer #2**
**An eleven year record of XCO$_2$ estimates derived from GOSAT measurements using the NASA ACOS version 9 retrieval algorithm**

T.E. Taylor et al.

Thank you to the reviewer for the suggestions and comments. We appreciate your time and concern. We have addressed each enumerated point below. The original reviewer comment is given in black. Our reply is given in blue. Modifications to the manuscript text are given in red as needed.

1. L28: "TANSO-FTS", the abbreviation is not introduced.

   It was introduced in the abstract, but I see the journal house rules require it to be introduced again in the main text. Therefore, the sentence now reads:

   Each day, GOSAT's Thermal And Near infrared Sensor for carbon Observation - Fourier Transform Spectrometer (TANSO-FTS) acquires approximately ten thousand high spectral resolution measurements of reflected sunlight ($\simeq 36.5$ M in ten years).

2. L30-31: Focus of the manuscript is ACOSv9 algorithm that does not deal with CH4. Is it necessary to mention XCH4?

   This is a valid point and we have removed the references to methane, here and at the opening of Section 2. We also removed the citation of [Parker, ESSD, 2020, GOSAT-CH4-ten-years].

3. L55: "the L2FP retrieval", do you mean L2FP retrieval algorithm?

   Correct.

   Motivated by these early studies, as well as the launch of the OCO-2 sensor in July 2014, the ACOS team continued to refine the L2FP retrieval algorithm.

4. L71: Add a comma after "In Section 3".

   Done.

5. L191: "IDP", the abbreviation is not introduced.

   The IDP was inadvertently introduced later in the document, so we moved the introduction to be at the first use instead of later.

6. L215: Why were these specific collocation criteria of +/- 2.5 ° Lat and +/-5 Lon selected? Also mentioned by the first referee.

To collocate GOSAT with TCCON, we followed the spatial criteria laid out in Section 4 of [Wunch, AMT, 2017, https://doi.org/10.5194/amt-10-2209-2017]. The sentence now reads: Following the criteria defined in (Wunch, 2017), the spatial collocation criteria for GOSAT soundings were those falling within $\pm 2.5°$ latitude and $\pm 5°$ longitude of a TCCON station for most sites. It is worth keeping in mind that GOSAT samples on any given orbit are order several hundred kilometers apart, i.e., it is quite spatially sparse. Therefore the spatial collocation criteria cannot be too strict.

7. L453: "ACOS GOSAT v9 XCO2 versus OCO-2", the comparative results can be summarized into a table for the convenience of the reader.

This is a great suggestion. We compiled a table listing the N, means delta $XCO_2$ and standard deviation by year and season for each observation mode. This helps provide a more complete picture of the differences between the two $XCO_2$ products. The following brief discussion and table were added to the text.

A set of summary statistics for the ACOS GOSAT v9 versus OCO-2 v10 $XCO_2$ product is given in Table 1. The values reported here are on the individual collocations by year and season, rather than the spatially gridded averages as given in Figures 13 and 14 [in the original submission]. For the land observations, there has been a very slight upward trend in time of the $\Delta XCO_2^{OCO\text{-}2}$ to slightly more positive values (GOSAT v9 larger than OCO-2 v10 $XCO_2$). On the other hand, for Ocean-Glint observations, the general trend has been an increasingly more negative $\Delta XCO_2^{OCO\text{-}2}$ in time, as was seen in Figure 15 [in the original submission]. Additional investigation will be required to determine the root cause(s) of these differences.

.

**Table 1.** A set of summary statistics for the comparison of the ACOS GOSAT v9 $XCO_2$ to the OCO-2 v10 product. Individual collocations for each year and season are given by N, while the mean $\Delta XCO_2$ and the standard deviation from the mean are given by $\mu$ and $\sigma$, respectively, both in units ppm. The top portion of the table is for land observations, while the bottom is for Ocean-Glint (OceanH).

| Land | DJF | | | MAM | | | JJA | | | SON | | |
|---|---|---|---|---|---|---|---|---|---|---|---|---|
| Year | N | $\mu$ | $\sigma$ | N | $\mu$ | $\sigma$ | N | $\mu$ | $\sigma$ | N | $\mu$ | $\sigma$ |
| 2014 | 0 | – | – | 0 | – | – | 0 | – | – | 4564 | 0.11 | 1.37 |
| 2015 | 1963 | 0.02 | 1.36 | 3160 | 0.15 | 1.47 | 5631 | -0.08 | 1.34 | 4108 | -0.01 | 1.39 |
| 2016 | 4379 | 0.01 | 1.28 | 3672 | -0.07 | 1.42 | 4701 | -0.04 | 1.41 | 4923 | 0.02 | 1.43 |
| 2017 | 3610 | 0.10 | 1.40 | 4097 | -0.04 | 1.37 | 3450 | 0.02 | 1.34 | 2892 | -0.02 | 1.41 |
| 2018 | 3605 | 0.11 | 1.41 | 3904 | -0.06 | 1.33 | 4738 | 0.06 | 1.36 | 4218 | 0.21 | 1.46 |
| 2019 | 2779 | 0.16 | 1.39 | 3917 | 0.06 | 1.41 | 4911 | 0.09 | 1.35 | 4823 | 0.24 | 1.40 |
| 2020 | 3422 | 0.17 | 1.37 | 3833 | -0.04 | 1.34 | 1183 | -0.03 | 1.22 | 0 | – | – |

| OceanH | DJF | | | MAM | | | JJA | | | SON | | |
|---|---|---|---|---|---|---|---|---|---|---|---|---|
| Year | N | $\mu$ | $\sigma$ | N | $\mu$ | $\sigma$ | N | $\mu$ | $\sigma$ | N | $\mu$ | $\sigma$ |
| 2014 | 0 | – | – | 0 | – | – | 0 | – | – | 2603 | 0.10 | 0.76 |
| 2015 | 2373 | 0.29 | 0.74 | 3139 | 0.14 | 0.84 | 2982 | -0.13 | 0.87 | 3103 | 0.14 | 0.81 |
| 2016 | 6029 | 0.07 | 0.80 | 5460 | -0.45 | 0.90 | 5109 | -0.66 | 0.89 | 7673 | -0.26 | 0.85 |
| 2017 | 6847 | -0.13 | 0.82 | 6235 | -0.39 | 0.87 | 3523 | -0.55 | 0.94 | 4332 | -0.18 | 0.85 |
| 2018 | 5888 | -0.05 | 0.86 | 5680 | -0.47 | 0.89 | 2767 | -0.65 | 0.90 | 5887 | -0.32 | 0.85 |
| 2019 | 4815 | -0.14 | 0.91 | 4887 | -0.50 | 0.93 | 4511 | -0.61 | 0.92 | 6972 | -0.37 | 0.95 |
| 2020 | 5451 | -0.34 | 0.94 | 4812 | -0.60 | 1.01 | 385 | -0.69 | 1.08 | 0 | – | – |

---

## Author Response (AR2)

**Response to Editor**
**An eleven year record of $XCO_2$ estimates derived from GOSAT measurements using the NASA ACOS version 9 retrieval algorithm**

T.E. Taylor et al.

Thank you to the editor for the suggestions and comments. We appreciate your time and concern. We have addressed each enumerated point below. The original reviewer comment is given in black. Our reply is given in blue. Modifications to the manuscript text are given in red as needed.

5     1. Requirement: Please include DOI information for the data products, listed at lines 907 and following in v3 of the manuscript, also at the end of the abstract. ESSD requires same information in both locations. Some readers, attracted by title and abstract, will choose to go directly to download options. Done.

The ACOS GOSAT v9 $XCO_2$ data are available on the NASA Goddard Earth Science Data and Information Services Center (GES-DISC) in both the per-orbit full format (https://doi.org/10.5067/OSGTIL9OV0PN) and in the per-day lite

10     format (https://doi.org/10.5067/VWSABTO7ZII4). In addition, a new set of monthly super-lite files, containing only the most essential variables for each satellite observation, has been generated to provide entry level users with a light weight satellite product for initial exploration (CaltechDATA, https://orcid.org/0000-0003-1080-9922). The v9 ACOS Data User's Guide (DUG) describes best-use practices for the GOSAT data. The GOSAT v9 data set should be especially useful for studies of carbon cycle phenomena that span a full decade or more, and may serve as a useful complement to

15     the shorter OCO-2 v10 data set, which begins in September 2014.

   2. Suggestion: Consider a 'teaser' data product. The GES DISC provides excellent access, including temporal and geographic sub-setting options, to both (L2Std and L2Lite) data products, but full downloads of either one involve 150 to 220 GB. (TCCON data require additional separate download).) Many interested users, not on high-bandwidth networks, will find such file sizes forbidding. Each landing page includes an example (global 1-year) graphic. Please may I sug-

20     gest a teaser file: a (geographically, temporally) small product that shows off your complete processing and analytical skills but that allows many users to download and evaluate before allocating time and resources for a full download. Many ESSD products, particularly those involving multi-year satellite records, use teaser products successfully. Ideally, a teaser carries its own DOI (mostly to protect you) and, like the data DOI, gets mentioned at end of the abstract as well as in a data availability section. Occasionally, teaser products reside at a separate data repository from the main data files,

25     e.g. Zenodo. Some authors devote a few sentences or a short paragraph to explanation of the teaser. Having looked again at graphics in your manuscript, I make an additional suggestion: provide a teaser product consisting of global GOSAT

CO2 compared to TCCON for the time period MAM 2014? This time period lies within your overlap period so you can highlight v9 improvements vs. v7.3. (You can also include relevant TCCON obs?) A separate ESSD Special Issue on emissions and air quality (e.g. https://essd.copernicus.org/articles/special_issue1100.html) uses exactly that time period

30    for global and regional products, in part to encourage data providers to share one product covering a mutually-agreed time period. Although focused on reactive gases, I see strong mutual benefit between that portion of the user community and your satellite-based expertise. Definitely consider a teaser; exact subset up to you.

This was an excellent suggestion, and we have generated a new "teaser" product that we call the "super-lite" files. It is aggregated on a monthly basis and contains only the bias corrected $XCO_2$ for the "good" quality soundings, along

35    with a hand full of necessary time and location variables. We agree that this will be a nice asset for early adopters of the GOSAT satellite record. The necessary citation is given in the Abstract as shown above, and in the Data Availability section at the end of the paper.